# STRUCTURED UNCERTAINTY GUIDED CLARIFICATION FOR LLM AGENTS

## ABSTRACT

LLM agents with tool-calling capabilities often fail when user instructions are ambiguous or incomplete, leading to incorrect invocations and task failures. Existing approaches operate in unstructured language spaces, generating clarifying questions through prompting strategies that lack principled criteria for determining which questions to ask and when to stop. We introduce a principled formulation of *structured uncertainty* that operates directly over tool parameters and their domains, cleanly separating specification uncertainty (what the user wants) from model uncertainty (what the LLM predicts). Our formulation uses Expected Value of Perfect Information (EVPI) to quantify the disambiguation value of each potential question, balanced against aspect-based cost modeling that prevents redundant questioning. We demonstrate the versatility of this formulation through two applications. First, SAGE-Agent uses structured uncertainty for inference-time question selection, achieving 7–39% higher coverage on ambiguous tasks while reducing clarification questions by 1.5–2.7× compared to strong prompting and uncertainty-based baselines. Second, we show that structured uncertainty provides effective training signals: uncertainty-guided reward modeling boosts When2Call accuracy from 36.5% to 65.2% (3B model) and 36.7% to 62.9% (7B model) through uncertainty-weighted GRPO training, demonstrating more sample-efficient reinforcement learning for tool-calling agents. To enable evaluation, we present *ClarifyBench*, the first multi-turn dynamic tool-calling disambiguation benchmark. Our results establish structured uncertainty as a principled framework that improves both inference-time interaction efficiency and training-time sample efficiency in tool-augmented agents.

## 1 INTRODUCTION

LLM Agents are AI systems that extend large language models (LLMs) with the ability to take real-world actions autonomously accumulate observations (Huang et al., 2024b). These agents often invoke external APIs and tools based on structured function definitions, enabling interaction with databases, web services, and software applications (Schick et al., 2023). These agents have been successfully deployed across diverse domains including travel planning, document processing, finance, vehicle control, and drug discovery (Xie et al., 2024; Mathur et al., 2024; Yu et al., 2024; Huang et al., 2024a; Liu et al., 2024). However, their effectiveness is fundamentally limited by ambiguous or incomplete

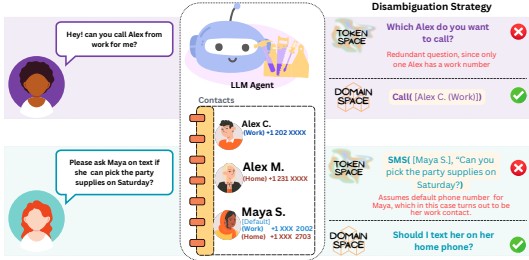

Figure 1: Linguistic-only disambiguation fails to use tool schemas, triggering unnecessary clarifications and inappropriate defaults. Grounding disambiguation in structured parameter domains avoids these problems.

user instructions that lead to incorrect tool invocations, failed transactions, and degraded user experience—problems that become increasingly critical as these systems handle more complex, high-stakes tasks. Ambiguity in user requests poses unique challenges for LLM agents, where imprecise interpretation can cascade into costly execution errors (Wang et al., 2024; Vijayvargiya et al., 2025). User ambiguity manifests through vague task specifications *("find me a good restaurant")*, incomplete parameters *("book a meeting for tomorrow")*, or implicit assumptions about system capabilities (Wang

et al., 2025). The structured nature of API schemas—with their specific parameter types, constraints, and interdependencies—amplifies this challenge, as a single ambiguous user query often maps to multiple valid API configurations with vastly different outcomes (Bandlamudi et al., 2025). For example, *"cancel my subscription"* could apply to multiple services, cancellation types (pause vs. permanent), or effective dates, each requiring different API calls with distinct consequences.

Existing disambiguation approaches suffer from fundamental limitations in the agentic tool-calling context. Due to their next-token prediction training, LLMs often hallucinate missing arguments when faced with incomplete information, leading to incorrect tool invocations (Wang et al., 2024). Current methods operate primarily in unstructured language spaces—generating clarifying questions as arbitrary text sequences through prompting strategies—rather than leveraging the structured constraints and dependencies that define tool schemas (Kobalczyk et al., 2025; Zhang et al., 2024). While prompting improvements can enhance question phrasing, they cannot fundamentally address the core limitation: without explicit modeling of parameter relationships, importance hierarchies, and feasibility constraints, agents lack principled criteria for determining which questions to ask and when to stop asking them. This results in over-clarification of low-impact details, under-clarification of critical missing information, and inability to distinguish feasible from infeasible requests, as demonstrated in Fig. 1. We address these limitations through a *structured uncertainty formulation* that operates directly in the space of tool parameters and their domains, rather than unstructured language space. By maintaining explicit probabilistic beliefs over structured tool-call candidates, our approach cleanly separates specification uncertainty (ambiguity in what the user wants) from model uncertainty (limitations in LLM capabilities). The key challenge is determining which clarifying question provides the most value—too many questions frustrate users, while too few lead to incorrect executions. We resolve this through Expected Value of Perfect Information (EVPI), a principle from Bayesian decision theory that quantifies how much each potential question would reduce uncertainty about the correct tool call in expectation.

**Contributions:** ➢ We introduce a principled formulation of *structured uncertainty* over tool-call parameters, using Expected Value of Perfect Information (EVPI) to optimally balance information gain against question cost through aspect-based redundancy modeling. This formulation cleanly separates specification uncertainty from model uncertainty by operating directly in the structured space of tool parameters and their domains. ➢ We demonstrate two applications of this formulation: **(i) SAGE-Agent**, which uses structured uncertainty for inference-time question selection, substantially improving task success rates while reducing clarification overhead compared to prompting and uncertainty-based baselines; and **(ii) uncertainty-guided reward modeling**, where structured uncertainty serves as an effective training signal to train tool-calling models. ➢ We present *ClarifyBench*, the first benchmark for multi-turn tool-calling disambiguation, equipped with an LLM-based user simulator supporting realistic conversational progression across diverse domains including document editing, vehicle control, stock trading, travel booking, and file system manipulation.

## 2 RELATED WORK

The challenge of resolving ambiguity in user interaction with LLMs through clarifying questions has gained increasing attention, particularly in tool-calling contexts. Early approaches to clarification focused on general dialogue systems, developing ranking-based methods for question selection (Rao & Daumé III, 2018; Xu et al., 2019) and Seq2Seq generation (Deng et al., 2022). Recent work has specifically addressed ambiguity in tool-calling scenarios: Ask-before-Plan introduces proactive planning agents that predict clarification needs and collect information before execution (Zhang et al., 2024), while Active Task Disambiguation frames the problem through Bayesian Experimental Design to maximize information gain from clarifying questions (Kobalczyk et al., 2025). Zhang and Choi propose intent-similarity based uncertainty estimation to determine when clarification is beneficial across various NLP tasks (Zhang & Choi, 2023). Complementary approaches explore training methods for clarification behavior: CollabLLM develops frameworks for transforming LLMs from passive responders into active collaborators (Wu et al., 2025), Zhang et al. teach LLMs to ask clarifying questions by modeling future conversation turns (Zhang et al., 2025), and Chen et al. propose action-based contrastive self-training for multi-turn clarification dialogues (Chen et al., 2025). Related efforts explore implicit intention understanding in language agents (Qian et al., 2024) and proactive dialogue systems that can handle ambiguous queries through goal planning (Deng et al.,

2023). However, these approaches primarily operate in the general language space without leveraging the structured nature of tool schemas.

## 3 THEORY

Modern LLM agents extend beyond text generation to become *agentic systems* that can interact with external tools and APIs to accomplish complex tasks. These agents typically follow a perception-reasoning-action cycle: they receive user queries, reason about appropriate actions, select and parameterize tool calls, and execute them to achieve desired outcomes. However, this paradigm faces a fundamental challenge when user queries are ambiguous or underspecified—the agent must somehow resolve uncertainty about both *which* tool to use and *how* to parameterize it.

### 3.1 STRUCTURED TOOL-CALLING AND BELIEF STATE

We model an LLM agent as a system $\mathcal{M}$ with access to a toolkit $\mathcal{T} = \{T_1, T_2, \ldots, T_K\}$. Each tool $T_i$ is characterized by a structured interface that defines its capabilities and parameter requirements.

**Definition 1 (Tool Schema).** A tool $T_i$ is defined by the tuple $(name_i, \Theta_i, \mathcal{D}_i, \mathcal{R}_i)$ where $name_i \in \mathbb{S}$ is the tool identifier, $\Theta_i = \{\theta_{i,1}, \ldots, \theta_{i,m_i}\}$ is the parameter set, $\mathcal{D}_i = \{\mathcal{D}_{i,1}, \ldots, \mathcal{D}_{i,m_i}\}$ with $\mathcal{D}_{i,j}$ the domain of $\theta_{i,j}$ i.e the set of allowed values, and $\mathcal{R}_i \subseteq \Theta_i$ specifies required parameters.

**Definition 2 (Tool Call Candidate).** A tool call candidate $c_i$ for tool $T_i$ is a partial function $c_i : \Theta_i \to \mathcal{D}_i \cup \{\bot\}$ where $c_i(\theta_{i,j}) = \bot$ indicates an unspecified parameter.

The agent's task is to map from an ambiguous natural language query $u$ to a fully specified tool call $c^* = (T^*, \boldsymbol{\theta}^*)$ where all required parameters are specified. The *candidate space* $\mathcal{C} = \{(T_i, c_i) : T_i \in \mathcal{T}, c_i \text{ is valid for } T_i\}$ represents all possible completions consistent with current information.

> 💡 **Uncertainty Quantification:** Methods that model uncertainty or disambiguation needs based on LLM response distributions must compute $p(\text{ambiguous}|u) = \sum_{\boldsymbol{w}} f(\boldsymbol{w}) p_{LLM}(\boldsymbol{w}|u)$ where $f$ determines if LLM response $\boldsymbol{w}$ indicates ambiguity. This conflates model uncertainty with specification uncertainty since the determination function $f$ itself depends on model capabilities. Our structured approach directly parameterizes $p(T_i, \boldsymbol{\theta}_i|u)$, cleanly separating these uncertainty sources.

**Definition 3 (Structured Belief State).** At time $t$, given the initial user query $u$ and accumulated responses $\{r_1, \ldots, r_t\}$, we maintain a belief distribution over the candidate space:

$$\mathcal{B}(t) = \{(c_i, \pi_i(t)) : c_i \in \mathcal{C}\}$$

where $\pi_i(t) \in [0, 1]$ represents the probability that candidate $c_i$ matches the user's true intent.

We decompose the joint probability as

$$p(T_i, \boldsymbol{\theta}_i \mid u, \{r_1, \ldots, r_t\}) = p(\boldsymbol{\theta}_i \mid T_i, u, \{r_1, \ldots, r_t\}) \, p(T_i \mid u)$$

and assume a uniform prior over tools $p(T_i \mid u) = 1/K$.[1]

Under a conditional independence assumption across parameters (for tractability), candidate probability becomes:

$$\pi_i(t) \propto \prod_{j=1}^{m_i} p(\theta_{i,j} \mid T_i, u, \{r_1, \ldots, r_t\})$$

where parameter certainty is $p(\theta_{i,j}) = 1$ if specified, $|\mathcal{D}_{i,j}(t)|^{-1}$ if unspecified with finite domain, and $\epsilon$ ($0 < \epsilon \ll 1$) for infinite/continuous domains. Here, $\mathcal{D}_{i,j}(t)$ is the feasible parameter domain after incorporating constraints from responses.

---

[1]This assumption reflects that, in practice, tools are proposed without strong prior bias. Future work could incorporate learned tool usage patterns or contextual priors.

**Belief Updates.** After asking question $q_t$ and receiving response $r_t$, beliefs update through domain constraint propagation:

$$\mathcal{D}_{i,j}(t+1) = \mathcal{D}_{i,j}(t) \cap \text{ExtractConstraints}(r_t, \theta_{i,j}, T_i) \tag{1}$$

$$\pi_i(t+1) \propto \pi_i(t) \cdot P(r_t|c_i, q_t) \cdot \prod_j p(\theta_{i,j}|T_i, u, \{r_1, \ldots, r_t\}) \tag{2}$$

### 3.2 INFORMATION-THEORETIC QUESTION SELECTION

The disambiguation process involves sequential decision-making: at each turn, the agent must decide whether to ask a clarifying question or execute the current best candidate. We formalize this decision through an information-theoretic criterion that balances information gain against question cost.

**Expected Value of Perfect Information.** Drawing from Bayesian decision theory and value of information frameworks (Rainforth et al., 2024), we quantify the expected benefit of asking question $q$ using the Expected Value of Perfect Information (EVPI).

**Definition 4 (Expected Value of Perfect Information).**

$$\text{EVPI}(q, \mathcal{B}(t)) = \mathbb{E}_{r \sim P(r|q, \mathcal{B}(t))} \left[ \max_{c_i \in \mathcal{C}} \pi_i(t|q, r) \right] - \max_{c_i \in \mathcal{C}} \pi_i(t) \tag{3}$$

where the response distribution is $P(r|q, \mathcal{B}(t)) = \sum_i \pi_i(t) P(r|c_i, q)$. EVPI naturally handles both tool disambiguation and parameter clarification in a unified framework—questions helping resolve tool choice and parameter values are evaluated using the same information-theoretic criterion.

**Aspects and Question Coverage.** We introduce **aspects** as the atomic unit of disambiguation. An aspect $a_{i,j}$ refers to parameter $\theta_{i,j}$ of tool $T_i$. The full set of aspects is

$$\mathcal{A} \triangleq \{a_{i,j} \mid i \in [1..K], \ j \in [1..m_i]\}.$$

A clarifying question targets a subset of aspects: for question $q$ we write $\mathcal{A}(q) \subseteq \mathcal{A}$. For bookkeeping we count how often an aspect has been targeted up to time $t$ as

$$n_a(t) \triangleq |\{\tau \le t : a \in \mathcal{A}(q_\tau)\}|.$$

**Definition 5 (Redundancy Cost).** Pure information maximization can lead to excessive questioning. We introduce a cost model that penalizes redundant questions about previously addressed aspects. For question $q$ targeting aspects $\mathcal{A}(q)$, with aspect history $n_a(t)$:

$$\text{Cost}(q, t) = \lambda \sum_{a \in \mathcal{A}(q)} n_a(t) \tag{4}$$

where $\lambda$ controls the penalty strength for redundant questions.

> 💡 **Structured Response Handling:** Past methods sample from $p(\text{solution}|q)$, requiring expensive enumeration. We treat responses as constraints $r \rightsquigarrow \mathcal{D}_{i,j}(t+1) = \mathcal{D}_{i,j}(t) \cap C(r)$ where $C(r)$ extracts constraints, enabling exact EVPI computation over finite patterns.

**Question Selection and Stopping Criteria.** At each timestep, we select the question that maximizes net information gain:

$$q^*(t) = \arg \max_{q \in \mathcal{Q}}[\text{EVPI}(q, \mathcal{B}(t)) - \text{Cost}(q, t)] \tag{5}$$

$$\text{Stop when: } \max_q[\text{EVPI}(q, \mathcal{B}(t)) - \text{Cost}(q, t)] < \alpha \cdot \max_i \pi_i(t) \tag{6}$$

This policy requires only one-step belief propagation for each candidate question, making it computationally tractable while maintaining principled information-theoretic grounding.

## 4 CLARIFYBENCH

The evaluation of clarification strategies in tool-calling agents requires benchmarks that capture the complexity of real-world user interactions, particularly when dealing with ambiguous or infeasible requests. As shown in Table 1, existing benchmarks exhibit critical limitations: many lack support for ambiguous and infeasible queries, while those that include such scenarios are limited in scope or domain coverage. Most critically, they rely on static evaluation without dynamic user simulation capabilities.

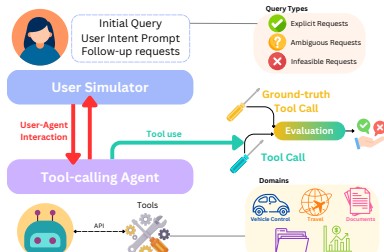

We introduce **ClarifyBench** to address these limitations. The task involves multi-turn interactions between a tool-equipped LLM agent and a user simulator that maintains the true user intention and responds to clarifying questions. The agent must identify when clarification is needed, pose appropriate questions, and execute correct tool calls based

Figure 2: ClarifyBench evaluates agent clarification strategies through multi-turn interactions between a user simulator and tool-equipped LLM agents across normal, ambiguous, and infeasible queries in 5 domains.

on the information gathered, while the simulator provides contextually relevant responses that guide the agent toward the intended action. As illustrated in Figure 2, **ClarifyBench** provides: (1) dynamic user simulation enabling natural conversational progression where users pose follow-up requests after clarification exchanges; (2) comprehensive coverage across three query types (normal, ambiguous, and infeasible); and (3) multi-domain evaluation spanning five distinct domains. Evaluation compares ground truth tool calls with agent-generated actions, providing robust assessment of clarification effectiveness across realistic scenarios.

| Benchmark | Dynamic User Simulation | Ambiguous Queries | Infeasible Queries | Multi-turn Requests | Tool Domains | Number of Tools |
|---|---|---|---|---|---|---|
| AgentBoard (Ma et al., 2024) | ✗ | ✗ | ✗ | ✗ | Information Retrieval, Manipulation | 50 |
| τ-bench (Yao et al., 2024) | ✓ | ✗ | ✗ | ✓ | Retail, Airlines | 24 |
| MMAU (Yin et al., 2024) | ✗ | ✗ | ✗ | ✗ | RapidAPI Tools | 364 |
| ToolSandbox (Lu et al., 2024) | ✓ | ✗ | ✗ | ✓ | Personal Assistant | 34 |
| Ask-Before-Plan (Zhang et al., 2024) | ✓ | ✓ | ✓ | ✗ | Travel | 6 |
| BFCL-v3 (Patil et al., 2025) | ✗ | ✓ | ✗ | ✓ | Vehicle Control, Stocks, Travel, File System | 129 |
| **ClarifyBench** | ✓ | ✓ | ✓ | ✓ | Documents, Vehicle Control, Stocks, Travel, File System | 92 |

Table 1: Comparison of ClarifyBench with existing tool-calling benchmarks.

## 4.1 BENCHMARK DESIGN

ClarifyBench encompasses five diverse domains that reflect real-world tool-calling scenarios: document processing, vehicle management, stock trading, travel planning, and file system management. These domains were selected to represent varying levels of complexity, different types of argument structures, and distinct sources of ambiguity that agents encounter in practice. Table 2 gives a statistical summary of the benchmark. Each sample in ClarifyBench is represented as a tuple: *(user query, user intent, follow-up queries, ground truth tool call, domain)*.

The benchmark includes three distinct query types that systematically evaluate different aspects of clarification: **1. Explicit Queries:** Well-specified requests that provide sufficient information for direct tool execution, serving as baseline performance indicators. **2. Ambiguous Queries:** Requests with missing or unclear parameters that require clarification to determine the appropriate tool calls and arguments. **3. Infeasible Queries:** Requests which if executed at face value would generate errors due to invalid parameters, conflicting constraints, or impossible conditions.

## 4.2 BENCHMARK CONSTRUCTION

**Data Sources.** ClarifyBench draws from two primary sources to ensure diversity and realism. First, we extract successfully executed tool calls from DocPilot (Mathur et al., 2024), which provides real user interactions in document processing scenarios. Second, we leverage the Berkeley Function Calling Leaderboard (BFCL-v3) (Patil et al., 2025), which offers data across multiple domains: vehicle control, stock trading, travel planning, and file system management.

**Data Augmentation.** To create the comprehensive set of query types required for clarification evaluation, we employ systematic data augmentation techniques. We process *DocPilot* dataset by anonymizing user metadata, replacing specific file names and domain terms in tool calls with

LLM-generated substitutes to ensure generalizability, followed by PII removal. For ambiguous queries, we randomly select upto 3 arguments from successful tool calls and obfuscate them, then prompt GPT-4o to generate five alternative user queries that omit the obfuscated information.For infeasible queries, we design handwritten rules based on common API errors to create tool calls that would generate failures, followed by a similar LLM-based query augmentation process. We process *BFCL-v3* using existing explicit and ambiguous parameter queries from the benchmark, ensuring sample independence by removing cases with secondary API dependencies. We apply rule-based validation and LLM judgment (via in-context learning) to identify and exclude such cases. For retained samples, we strip secondary API utterances and tool calls from ground truth annotations. User intent prompts are generated through LLM based detailed summarization of the ground truth tool calls and user utterances.

| Metric | Doc | Vehicle | Stocks | Travel | Files | All |
|---|---|---|---|---|---|---|
| Total Samples | 181 | 139 | 143 | 119 | 134 | 716 |
| Number of Tools | 18 | 22 | 19 | 15 | 18 | 92 |
| Avg # of Tool Calls | 3.9 | 4.5 | 3.9 | 3.7 | 3.1 | 3.8 |
| Explicit Queries | 49 | 50 | 49 | 50 | 43 | 241 |
| Ambiguous Queries | 49 | 39 | 46 | 40 | 39 | 213 |
| Infeasible Queries | 48 | 49 | 38 | 18 | 45 | 198 |
| Avg # of Follow-up | 2.9 | 2.1 | 2.7 | 2.3 | 1.8 | 2.4 |

Table 2: Statistical description of ClarifyBench.

**Human Validation.** To ensure quality and naturalness, a human annotator evaluates all LLM-generated queries using three criteria: (A) naturalness of language, (B) faithfulness to the expected tool calls with all required details and no obfuscated parameters, and (C) for infeasible queries, the presence of explicit error-inducing requirements. Two annotators assign a 5-point Likert score to every candidate query, and the final selected query for a sample is the one that receives the highest score. Inter-annotator agreement for the highest-scoring selections is given by Cohen's $\kappa = 0.76$.

# 5 STRUCTURED ARGUMENT UNCERTAINTY GUIDED ELICITATION AGENT

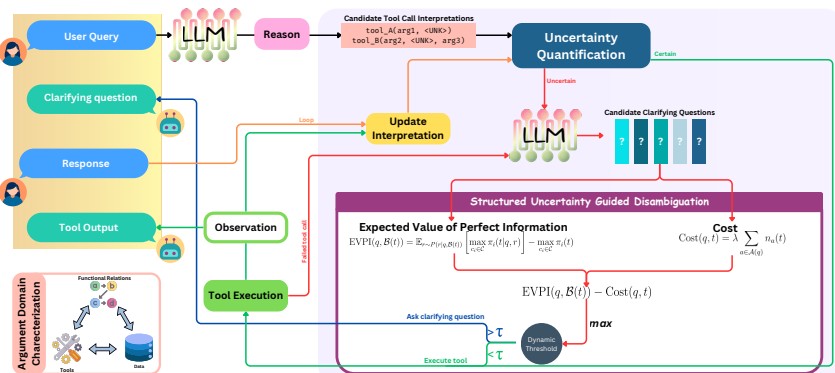

Figure 3: **SAGE-Agent**: ❶) Given a user query, an LLM reasons and generates potential tool calls with possibly uncertain parameters. These tool calls undergo (❷) structured uncertainty quantification to determine if clarification is needed. When uncertainty exists, the agent uses an LLM to produce (❸) candidate clarifying questions, and scores them using (❹) a cost-penalized Eexpected Value of Perfect Information (EVPI) metric. Tool-parameter domain interpretation is updated based on user-response to the clarifying question (❺), and given no further uncertainty, the best tool call is executed ❻.

SAGE (Structured Argument Uncertainty guided Elicitation) augments the standard Reason–Act–Observe loop by inserting structured, domain-aware clarification into the *Reason* stage (as seen in Fig. 3). Let the user input be $u$; the toolkit $\mathcal{T}$ and tool schemas follow Definition 1.

## 5.1 AGENT FLOW

At step $t$, the agent maintains belief $\boldsymbol{\pi}(t) = \{\pi_c(t)\}_{c\in\mathcal{C}}$ and observations $\mathcal{O}_t$. The full loop can be written as a combination of Reason ($\mathcal{R}$) and Act ($\mathcal{A}ct$):

$$\left(\mathcal{C}_t, Q_t\right) \xleftarrow{\mathcal{R}} (u, \mathcal{O}_t, \mathcal{T}) \xrightarrow{\mathcal{A}ct} a_t = \begin{cases} \text{execute}: & c^*(t) = \arg\max_c \pi_c(t) \\ q^*: & \boldsymbol{\pi}(t+1) = \mathcal{O}b(\boldsymbol{\pi}(t), o_{t+1}) \end{cases}$$

where $\mathcal{R}$ produces candidate tool calls $\mathcal{C}_t$ and aspect-targeted questions $Q_t$, $\mathcal{A}ct$ selects either execution or query, and $\mathcal{O}b$ performs domain-constrained belief refinement (Fig. 3).

## 5.2 Candidate Generation, Questioning, and Belief Update

At step $t$, SAGE proceeds as follows:

**1. Candidate Generation.** The **Reason** stage prompts an LLM with $(u, \mathcal{O}_t, \mathcal{T})$ to produce candidate tool calls $\mathcal{C}_t = \{c_1, \ldots, c_N\}$, each assigning parameters $\Theta_{i(c)}$ concrete values or `<UNK>`. Candidate certainty is defined as $\pi_c(t) = \prod_{\theta_{i,j} \in \Theta_{i(c)}} p(\theta_{i,j} \mid T_{i(c)}, \text{obs}_t)$. If $\max_c \pi_c(t) \geq \tau_{\text{exec}}$, execute $c^*(t) = \arg\max_c \pi_c(t)$; otherwise continue.

**2. Question Generation.** An LLM is prompted with (i) $q$, (ii) $\mathcal{C}$ and masks, (iii) tool schemas, and (iv) recent observations to output $Q = \{(q_k, c_{i_k}, A_k)\}_{k=1}^L$, where $q_k$ is the question text, $c_{i_k}$ the candidate being disambiguated, and $A_k \subseteq \mathcal{A}$ the targeted aspects (parameters). Output is machine-parsable with `<UNK>` for ambiguous parameters.

**3. Scoring and Selection.** Let $\mathcal{P}_q = \{C_1, \ldots, C_M\}$ be the partition of $\mathcal{C}_t$ induced by $A$. The EVPI is $\text{EVPI}(q) = \sum_{m=1}^M \max_{c \in C_m} \pi_c(t) - \max_{c \in \mathcal{C}_t} \pi_c(t)$. Score each question as $\text{Score}(q,t) = \text{EVPI}(q) - \lambda \sum_{a \in A} n_a(t)$, select $q^*(t) = \arg\max_q \text{Score}(q,t)$. If $\max_q \text{Score}(q,t) < \alpha \max_c \pi_c(t)$ or budget $n_s$ is exhausted, execute $c^*(t)$.

**4. Belief Update.** After observing answer $r$, update domains as $\mathcal{D}_{i,j}(t+1) \leftarrow \mathcal{D}_{i,j}(t) \cap f_{\text{update}}(\theta_{i,j}, r)$ and recompute $\pi_c(t+1)$.

**5. Termination & Error Recovery.** Stop if (i) $\max_c \pi_c(t) \geq \tau_{\text{exec}}$, (ii) $\max_q \text{Score}(q,t) < \alpha \max_c \pi_c(t)$, or (iii) $t \geq n_s$. On execution failure, prompt for a fix or generate an error-specific $q_{\text{error}}$ and re-enter step 3.

## 6 Reward Modeling with Structured Uncertainty

Our objective is to teach the agent not only *what* action to take but *when* to act with confidence versus request clarification. We fine-tune the policy using **Group Relative Policy Optimization (GRPO)** (Shao et al., 2024), which samples multiple candidate actions per prompt, computes relative rewards, and updates the policy towards those exceeding the group mean—yielding a critic-free, memory-efficient variant of PPO that stabilizes optimization through implicit baselining and KL regularization. Our training data comes from the 9K examples in the `When2Call` (Ross et al., 2025) dataset. For each user prompt and its tool set, the agent may take exactly one of four actions: `AskQuestion`, `CallTool(parameters)`, `Decline`, or `DirectAnswer`. We prompt a base model to emit structured tags `<reason>. . . </reason> <answer>...</answer>`, and from that we compute scalar rewards.

### 6.1 Baseline Reward

The baseline reward is $r_{\text{base}} = r_{\text{fmt}} + r_{\text{tool}} + r_{\text{cls}}$, where $r_{\text{fmt}} = 1.5$ (correct schema), $r_{\text{tool}}$ equals $1.0$ for correct tool+parameters, $0.75$ if tool is correct but parameters are wrong, and $0.5$ for correctly identifying a tool call or for non-tool actions, and $r_{\text{cls}}$ equals up to $2.0$ for correct action type. This encourages correctness and well-formedness but treats all instantiations equally regardless of model confidence or question informativeness.

### 6.2 Certainty-Weighted Reward (Ours)

Let $\pi_c(t)$ be the belief over candidate tool calls $c \in \mathcal{C}_t$. We define $\text{Cert}(a_t) = \max_c \pi_c(t)$ if $a_t$ is a tool call, $1 - \max_c \pi_c(t)$ if $a_t$ is a question, and $1$ otherwise. The category reward becomes $R_{\text{category}}(a_t) = \text{Cert}(a_t) \cdot r_{\text{base}}(a_t)$ which up-weights confident correct tool calls, penalizes low-certainty calls, and rewards clarification only when uncertainty is high—thus aligning reward with the agent's own epistemic state.

> 💡 **Key Insight:** Our reward is *self-calibrating*: it needs no critic to judge question quality, yet drives informative clarifications and confident tool calls. Unlike the baseline, which rewards all correct calls equally, our certainty-weighted reward **scales** with belief: confident calls get full payoff, low-confidence calls are penalized, and clarifications are rewarded only when uncertainty is high.

# 7 EXPERIMENTS

**(A) Agent Inference Experiment.** *1. ClarifyBench.* All baselines are implemented on a common ReAct agent scaffold for fair comparison. We evaluate **SAGE-Agent** against four baselines: (i) **ReAct + ask_question()**, a standard ReAct agent with an ask_question() tool serving as our control baseline; (ii) **ProCOT** (Deng et al., 2023), which performs ProActive Chain-of-Thought reasoning to anticipate ambiguities before tool use; (iii) **Active Task Disambiguation** (Kobalczyk et al., 2025), which generates candidate interpretations and clarification queries based on response entropy by parametrizing the solution space; and (iv) **Domain-aware ReAct**, which augments prompting and question generation with explicit schema information provided as context. All methods use GPT-4o and Qwen2.5-14B-Instruct with temperature $0.5$. For SAGE-Agent, we pick $\lambda = 0.5, \alpha = 0.1, \epsilon = 10^{-4}$. We evaluate using four metrics: (1) **Coverage Rate**: proportion of tool calls with correct parameters matching the ground truth; (2) **Tool Match Rate (TMR)**: tool match rate against ground truth; (3) **Parameter Match Rate (PMR)**: paramater match rate against ground-truth; and (4) **Average Number of Questions (#Q)**: mean number of clarification questions asked per task (lower is better). *2. BFCLv2 (When2Call)* We use the open-ended evaluation split of When2Call, built on top of BFCLv2 to perform single-turn validation of our method. We compared our method against a ReAct baseline and Active-task-Disambiguation, since this is single-turn validation and these baselines are representative of different disambiguation strategies. We used 2xRTXA600 for inference. **(B) Reward Modeling Experiment.** We trained GRPO with Qwen2.5-Instruct (3B and 7B) for one epoch using Unsloth (Daniel Han & team, 2023). Three independent runs were performed, and results from the best-performing model are reported. Evaluation follows the original paper: log-probability comparison across options, option-prompted selection, and direct prompting without options. We trained on 4xL40S GPUs, and inferred on 1xL40S GPU. We train each setting for 3 runs, and report the setting with the best results.

# 8 RESULTS

## 8.1 AGENT INFERENCE EXPERIMENTS

| Method | ClarifyBench - Ambiguous | | | | ClarifyBench - Explicit | | | | ClarifyBench - Infeasible | | | |
|---|---|---|---|---|---|---|---|---|---|---|---|---|
| | Coverage↑ | TMR↑ | PMR↑ | Avg #Q↓ | Coverage↑ | TMR↑ | PMR↑ | Avg #Q↓ | Coverage↑ | TMR↑ | PMR↑ | Avg #Q↓ |
| *Base LLM: GPT-4o* | | | | | | | | | | | | |
| ReAct + ask_question() | 42.88±25.1 | 70.41±27.3 | 62.55±23.9 | 2.68±2.4 | 61.17±22.7 | 87.95±25.8 | 71.99±28.4 | 2.15±2.7 | 58.85±24.3 | 85.05±26.1 | 75.09±21.8 | 2.21±2.6 |
| ProCOT | 54.27±27.4 | 75.62±29.1 | 66.82±24.6 | 2.07±2.2 | 66.98±22.8 | 89.57±28.7 | 72.80±25.4 | 2.14±2.5 | 61.48±24.2 | 89.32±27.5 | 74.41±23.5 | 2.43±2.8 |
| Active Task Disambiguation | 45.60±26.7 | 77.10±28.2 | 60.78±22.4 | 3.42±2.6 | 66.97±21.9 | 90.47±29.3 | 72.45±24.9 | 2.94±2.5 | 65.27±23.6 | 89.18±28.8 | 75.09±23.0 | 2.63±2.3 |
| Domain-aware ReAct | 55.70±24.5 | 79.83±25.7 | 68.04±23.3 | 2.56±2.1 | 68.11±22.5 | 91.17±26.1 | 74.04±25.2 | 2.10±2.6 | 61.48±24.0 | 90.32±25.4 | 76.46±26.7 | 2.03±2.7 |
| SAGE-Agent (Ours) Heuristic-based | 56.42±24.3 | 82.31±26.8 | 69.81±24.7 | 1.82±2.3 | 70.41±24.7 | 91.65±27.4 | 74.89±25.8 | **1.07**±2.4 | 66.23±23.9 | 90.52±26.5 | 76.64±25.3 | 1.48±2.5 |
| SAGE-Agent (Ours) | **59.73**±22.1 | **86.02**±27.5 | **71.79**±25.3 | **1.39**±2.0 | **71.67**±21.8 | **93.65**±29.7 | **75.94**±26.1 | 1.08±2.2 | **67.33**±23.4 | **92.89**±28.3 | **77.41**±27.9 | **1.26**±2.1 |
| *Base LLM: Qwen2.5-14B-Instruct* | | | | | | | | | | | | |
| ReAct + ask_question() | 40.34±33.9 | 68.92±32.0 | 63.35±31.5 | 1.78±1.94 | 51.85±33.8 | 89.20±22.8 | 73.63±28.9 | 1.69±1.67 | 42.39±32.4 | 70.82±31.1 | 63.31±34.0 | 1.82±1.43 |
| ProCOT | 52.45±33.5 | 71.78±33.7 | 70.08±33.2 | 1.89±2.03 | 61.76±31.5 | 84.08±23.8 | 74.60±28.4 | 1.69±1.68 | 52.08±31.4 | 71.92±29.3 | 68.72±35.0 | 1.78±1.51 |
| Active Task Disambiguation | 43.04±29.2 | 69.06±33.0 | 57.49±34.1 | 2.45±1.72 | 59.83±33.1 | 81.01±26.6 | 68.69±31.5 | 2.31±2.29 | 52.20±30.6 | 76.59±32.5 | 69.45±35.0 | 2.22±2.12 |
| Domain-aware ReAct | 51.10±31.9 | 75.31±30.7 | 67.50±31.5 | 2.07±1.35 | 60.91±34.2 | 86.91±24.8 | 71.70±28.7 | 1.61±1.56 | 55.76±31.7 | 81.06±27.2 | 72.23±32.0 | 1.66±1.30 |
| SAGE-Agent (Ours) Heuristic-based | 51.62±32.5 | **78.23**±30.9 | 74.03±31.8 | 1.67±1.85 | 62.45±33.4 | 89.89±23.2 | 73.89±29.1 | 1.23±1.74 | 59.88±31.2 | 84.12±28.6 | 75.51±32.8 | 1.75±1.62 |
| SAGE-Agent (Ours) | **54.56**±33.0 | 78.14±30.5 | **74.21**±32.2 | **1.41**±2.19 | **64.62**±33.6 | **92.05**±20.8 | **75.50**±28.2 | **0.93**±1.93 | **61.84**±30.8 | **85.26**±24.5 | **76.52**±29.5 | **1.49**±0.95 |

Table 3: Performance comparison of agent strategies on ClarifyBench across two base LLMs (GPT-4o and Qwen2.5-14B-Instruct). Best results within each LLM group are highlighted in bold.

**Performance Gains Across Task Categories.** On Ambiguous tasks with GPT-4o, SAGE-Agent achieves 59.73% Coverage Rate, substantially outperforming Domain-aware ReAct (55.70%), Pro-COT (54.12%), and basic ReAct (52.34%). This 4.03pp improvement over the strongest baseline extends to downstream metrics: Tool Match Rate reaches 86.02% versus 79.83% (Domain-aware ReAct) and 76.45% (basic ReAct), while Parameter Match Rate attains 71.79% versus 68.04% and 65.21% respectively. The pattern persists across Explicit scenarios, where SAGE-Agent achieves

71.67% Coverage (+3.56pp over Domain-aware ReAct, +5.23pp over basic ReAct), 93.65% TMR (+2.48pp, +4.12pp), and 75.94% PMR (+1.90pp, +3.67pp). Even on Infeasible tasks—where systems must recognize unsatisfiable queries, SAGE-Agent excels with 67.33% Coverage and 92.89% TMR, significantly outperforming Domain-aware ReAct (63.21%, 88.45%) and all other baselines. These results demonstrate that structured schema-based reasoning enables more accurate task interpretation than unstructured clarification approaches.

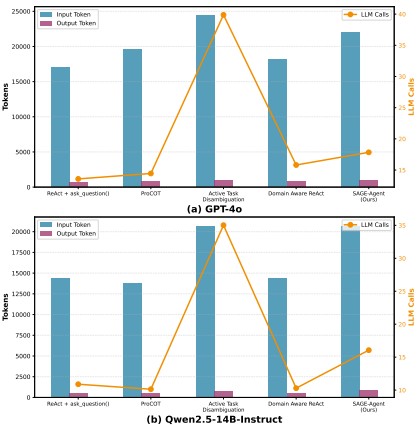

Figure 4: Resource consumption across methods for GPT-4o and Qwen2.5-14B.

**Dramatic Reduction in User Burden.** SAGE-Agent achieves superior performance while asking dramatically fewer questions. On Ambiguous tasks with GPT-4o, it averages just 1.39 questions per task; a 45.7% reduction versus Domain-aware ReAct (2.56 questions), 48.1% reduction versus basic ReAct (2.68 questions), and 59.4% reduction versus Active Task Disambiguation (3.42 questions). On Explicit scenarios where all information is present initially, SAGE-Agent asks only 1.08 questions, where all baselines should ideally approach 0.

**Computational Efficiency Despite Structured Reasoning.** Figure 4 reveals expected trade-offs: simpler baselines (ReAct, ProCOT, Domain-aware ReAct) use 14-18K tokens and 14-16 calls but sacrifice performance (Table 3). Among uncertainty-modeling methods, Active Task Disambiguation computes entropy over a |questions| × |solutions| matrix, requiring 24K tokens and 40 calls. SAGE-Agent instead parametrizes uncertainty directly over schema spaces, avoiding solution sampling entirely. This yields 22K tokens with 54% fewer API calls, reducing latency and cost while maintaining superior performance.

**Robustness Across Language Models.** SAGE-Agent's advantages generalize across both proprietary and open-source LLMs. With Qwen2.5-14B-Instruct, SAGE-Agent achieves 54.56% Coverage on Ambiguous tasks, outperforming ProCOT (52.45%) and Domain-aware ReAct (51.10%), while reducing questions from 2.07 to 1.41. While absolute metrics are lower with smaller models, relative improvements over baselines remain consistent, demonstrating systematic advantages independent of model choice.

**Ablation.** SAGE-Agent Heuristic Based is an ablation where questions are triggered by the presence of <UNK> tokens in tool calls, without using EVPI for question selection. This variant shows small but consistent performance degradation, ranging from 1-3 points across most metrics while asking 0.2-0.4 more questions on average. The heuristic approach triggers questions but lacks effective discrimination between them, and unlike the full system, it cannot resort to default execution when questions have low information value. These issues compound across the multi-turn ClarifyBench evaluation, leading to cumulative metric reductions.

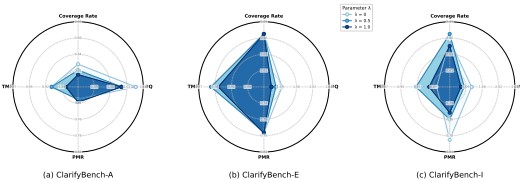

**Impact of $\lambda$.** The redundancy penalty weight $\lambda$ (Definition 5) controls the trade-off between information gathering and user burden by penalizing questions targeting previously queried aspects. Figure 5 shows the effect of $\lambda \in$

Figure 5: Effect of $\lambda$ on performance metrics across ClarifyBench splits. Increasing $\lambda$ from 0 to 0.5 reduces #Q by 18-27% while maintaining stable Coverage, TMR, and PMR ($< 3\%$ deviation).

$\{0, 0.5, 1.0\}$ across 70 samples from each ClarifyBench split using GPT-4o, with independently scaled radar axes. Increasing $\lambda$ from 0 to 0.5 yields substantial question reductions—18.1% on Ambiguous, 26.6% on Explicit, and 24.2% on Infeasible splits—while preserving task execution quality. Coverage Rate, TMR, and PMR remain stable with deviations under 3% across all settings, indicating that the penalized questions were indeed redundant rather than essential for task completion. The radar plots visualize this trade-off: the #Q dimension contracts inward while other metrics maintain consistent polygon shapes, demonstrating that question economy can be achieved without sacrificing accuracy.

**Single-Turn Disambiguation Performance**  Table 4 presents performance comparison on BFCLv2 When2Call. ReAct demonstrates high ToolCall recall (0.79) but exhibits poor Decline behavior (0.58 recall), indicating a bias toward tool invocation even for inappropriate requests. Active Task Disambiguation achieves high AskQuestion recall (0.74-0.78) but suffers from low precision (0.45-0.35), reflecting excessive questioning behavior. In contrast, SAGE-Agent achieves the best balance with highest ToolCall precision (0.80) while maintaining strong Decline performance (0.78 F1). Notably, these behavioral patterns persist across model scales from GPT-4o to Qwen2.5-14B-

| Method | ToolCall | | | AskQuestion | | | Decline | | |
|---|---|---|---|---|---|---|---|---|---|
| | P | R | F1 | P | R | F1 | P | R | F1 |
| *Base LLM: GPT-4o* | | | | | | | | | |
| ReAct | 0.71 | 0.79 | **0.75** | 0.59 | 0.69 | 0.64 | 0.87 | 0.58 | 0.69 |
| Act. Task Dis. | 0.61 | 0.24 | 0.34 | 0.45 | 0.74 | 0.56 | 0.74 | 0.73 | 0.73 |
| SAGE-Agent | 0.80 | 0.55 | 0.65 | 0.61 | 0.70 | **0.65** | 0.72 | 0.84 | **0.78** |
| *Base LLM: Qwen2.5-14B-Instruct* | | | | | | | | | |
| ReAct | 0.62 | 0.85 | **0.72** | 0.50 | 0.65 | 0.57 | 0.88 | 0.39 | 0.54 |
| Act. Task Dis. | 0.36 | 0.12 | 0.18 | 0.35 | 0.78 | 0.48 | 0.62 | 0.28 | 0.39 |
| SAGE-Agent | 0.76 | 0.48 | 0.59 | 0.53 | 0.75 | **0.62** | 0.79 | 0.76 | **0.77** |

Table 4: Performance comparison of agent strategies on BFCLv2 (When2Call).

Instruct, though with degraded absolute performance, suggesting that SAGE-Agent's structured approach provides more robust guidance for disambiguation decisions.

## 8.2 REWARD MODELING EXPERIMENTS

Figure 6 validates our hypothesis that uncertainty-aware training signals improve LLM clarification behavior. The When2Call benchmark tests models' ability to recognize when clarification is needed versus when to proceed with available information.

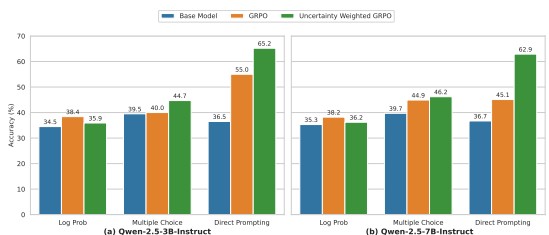

Figure 6: Performance of Qwen-2.5 models on When2Call across three evaluation methods: Log Probability, Multiple Choice, and Direct Prompting.

**Training Signal Impact.**  Base models without clarification training achieve poor performance (34.5–39.7% accuracy), demonstrating that recognizing clarification needs is non-trivial. Standard GRPO provides modest improvements, while uncertainty-weighted GRPO yields substantial gains (up to +28.7 percentage points). This validates that structured uncertainty measures provide more effective training signals than binary success/failure rewards.

**Model Scale vs. Signal Quality.** Comparing Qwen-2.5-3B and 7B models reveals that training signal quality matters more than model scale. The 3B model with uncertainty-weighted training (65.2% accuracy) substantially outperforms the 7B model with standard training (45.1% accuracy). This suggests that incorporating structured uncertainty into training objectives may be more valuable than simply scaling model parameters.

**Evaluation Mode Analysis.** The largest improvements occur in Direct Prompting mode, where models must make clarification decisions based solely on query analysis without multiple-choice scaffolding. This indicates that uncertainty-weighted training helps models develop robust internal representations of when clarification is needed, rather than merely improving selection among provided options.

## 9 CONCLUSION

Ambiguous user instructions fundamentally challenge tool-augmented LLM agents, leading to incorrect invocations and task failures. We presented **SAGE-Agent**, which models joint tool-argument clarification as a POMDP with Bayesian Value of Information objectives for optimal question selection. Extensive experiments validate our structured uncertainty approach: SAGE-Agent improves coverage on ambiguous tasks by 7–39% while reducing questions by 1.5–2.7× on *ClarifyBench*, and uncertainty-weighted GRPO training boosts *When2Call* accuracy from 36.5% to 65.2% (3B) and 36.7% to 62.9% (7B). These results demonstrate that structured uncertainty provides a principled foundation for both inference and learning in tool-augmented scenarios. Our work establishes structured uncertainty quantification as essential for reliable, efficient LLM agents in real-world applications.

## 10 ETHICS STATEMENT

Our research does not use any personally identifiable information (PII) and all datasets employed in this work are used in accordance with their respective licenses (Apache 2.0). Our paper is designed primarily for deployment in collaborative AI assistance contexts where resolving ambiguity enhances productivity and user experience while minimizing unnecessary interaction. The system's core approach of reducing clarification questions through principled uncertainty estimation promotes more equitable access to AI assistance by respecting users' time and cognitive resources. While SAGE-Agent significantly reduces interaction burden, we recommend appropriate transparency about system limitations and human oversight when deploying in sensitive contexts. Furthermore, we encourage ongoing evaluation to ensure that question selection patterns do not reflect or amplify biases present in underlying models or training data. We acknowledge the ICLR code of ethics.

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

# Appendices

## A    SAGE-AGENT

### A.1    THEORETICAL PROOFS

**Proposition 1 (Viability Score Properties).** *The viability scoring function satisfies: (1) Monotonicity:* $\pi_i(t+1) \geq \pi_i(t)$ *when information is gained, (2) Boundedness:* $0 \leq \pi_i(t) \leq 1$, *(3) Completeness:* $\pi_i(t) = 1$ *iff all parameters are fully specified.*

*Proof.* (1) **Monotonicity:** Information gain can only constrain parameter domains: $\mathcal{D}_{i,j}(t+1) \subseteq \mathcal{D}_{i,j}(t)$. Therefore $|\mathcal{D}_{i,j}(t+1)| \leq |\mathcal{D}_{i,j}(t)|$, which implies $|\mathcal{D}_{i,j}(t+1)|^{-1} \geq |\mathcal{D}_{i,j}(t)|^{-1}$. Since $\pi_i(t) = \prod_j p(\theta_{i,j})$ and each factor is non-decreasing, $\pi_i(t+1) \geq \pi_i(t)$.

(2) **Boundedness:** Each parameter certainty $p(\theta_{i,j}) \leq 1$ by definition. Since $\pi_i(t) = \prod_j p(\theta_{i,j})$, we have $0 \leq \pi_i(t) \leq 1$.

(3) **Completeness:** $\pi_i(t) = 1 \Leftrightarrow \prod_j p(\theta_{i,j}) = 1 \Leftrightarrow \forall j : p(\theta_{i,j}) = 1 \Leftrightarrow$ all parameters specified. $\square$

**Proposition 2 (EVPI Properties).** *The EVPI function satisfies: (1) Non-negativity:* $EVPI(q, \mathcal{B}(t)) \geq 0$, *(2) Submodularity: diminishing returns for question sequences, (3) Convergence: EVPI approaches zero as uncertainty resolves.*

*Proof.* (1) **Non-negativity:** By Jensen's inequality applied to the concave maximum function:

$$\mathbb{E}_r\left[\max_{c_i} \pi_i(t|q,r)\right] \geq \max_{c_i} \mathbb{E}_r[\pi_i(t|q,r)] = \max_{c_i} \pi_i(t)$$

Therefore $EVPI(q, \mathcal{B}(t)) \geq 0$.

(2) **Submodularity:** For question sets $S \subseteq S'$, the marginal information gain satisfies:

$$EVPI(q|S) - EVPI(q|S') = H[\mathcal{B}|S] - H[\mathcal{B}|S \cup \{q\}] - (H[\mathcal{B}|S'] - H[\mathcal{B}|S' \cup \{q\}]) \geq 0$$

This follows from submodularity of entropy: $H[X|Y] - H[X|Y,Z] \geq H[X|Y,W] - H[X|Y,W,Z]$ when $W \supseteq \emptyset$.

(3) **Convergence:** As uncertainty resolves, $\max_i \pi_i(t) \to 1$ and candidate distributions become concentrated. For any question $q$, $\mathbb{E}_r[\max_i \pi_i(t|q,r)] \to \max_i \pi_i(t)$, so $EVPI(q) \to 0$. $\square$

**Theorem 1 (Finite Termination).** *Under regularity conditions on the response model, the algorithm terminates in finite expected time with probability 1.*

*Proof.* The termination condition is $\max_q[EVPI(q) - Cost(q)] < \alpha \cdot \max_i \pi_i(t)$.

**Case 1:** If $\max_i \pi_i(t)$ increases over time (candidates improve), the right-hand side grows while EVPI values are bounded above. Eventually the inequality is satisfied.

**Case 2:** If $\max_i \pi_i(t)$ remains bounded, then either: - EVPI values decrease due to information gain (Proposition 2.3) while costs increase linearly - Or no informative questions remain, making EVPI $\approx 0$

In both cases, the net value becomes negative in finite time.

**Formal bound:** Let $\rho = \mathbb{E}[\text{improvement in } \max_i \pi_i \text{ per question}]$ and $\gamma = \mathbb{E}[\text{EVPI decline per question}]$. - If $\rho > 0$: termination when $\alpha\rho T \geq EVPI_{\text{initial}} - \gamma T$, giving $T \leq \frac{EVPI_{\text{initial}}}{\alpha\rho + \gamma}$ - If $\rho \leq 0$: termination when costs exceed EVPI, giving $T \leq \frac{\max EVPI}{\lambda \cdot \min |\mathcal{A}(q)|}$

Therefore $\mathbb{E}[T] < \infty$. $\square$

### A.2    COMPLETE ALGORITHM SPECIFICATION

**Algorithm.** Algorithm 1 presents the complete SAGE-Agent procedure. The algorithm maintains beliefs $\boldsymbol{\pi}(t)$ over candidate tool calls and aspect history $n_a(t)$ to track redundant questioning. At each timestep, the agent generates candidates via the reasoning stage $\mathcal{R}$ (line 6), computes viability scores (line 9), and checks if uncertainty exceeds threshold $\tau$ (line 12).

When uncertainty is high, the agent generates clarifying questions with their targeted aspects simultaneously (line 14), computes EVPI and redundancy costs (lines 17-21), and applies the stopping

---

**Algorithm 1** SAGE-Agent

---

**Require:** User query $u$, toolkit $\mathcal{T}$, max steps $T_{\max}$, redundancy penalty $\lambda$, stopping threshold $\alpha$, uncertainty threshold $\tau$

1: Initialize beliefs $\boldsymbol{\pi}(0) = \{\pi_c(0)\}_{c \in \mathcal{C}}$, observations $\mathcal{O}_0 = \emptyset$
2: Initialize aspect history $n_a(0) = 0$ for all $a \in \mathcal{A}$
3: **for** $t = 0, 1, \ldots, T_{\max}$ **do**
4:     **// Reason Stage** $\mathcal{R}$
5:     $\mathcal{C}_t \leftarrow \mathcal{R}(u, \mathcal{O}_t, \mathcal{T})$                           $\triangleright$ Generate candidate tool calls
6:
7:     **// Structured Uncertainty Quantification**
8:     Compute beliefs $\pi_i(t)$ for each $c_i \in \mathcal{C}_t$
9:     Compute uncertainty $U(t) = \max_{c_i \in \mathcal{C}_t} U(c_i)$
10:
11:     **if** $U(t) > \tau$ **then**                       $\triangleright$ Uncertainty exceeds threshold
12:         **// Generate Questions with Targeted Aspects**
13:         $\{(q, \mathcal{A}(q))\} \leftarrow \text{GenerateQuestions}(\mathcal{C}_t, u, \mathcal{O}_t, \mathcal{T})$    $\triangleright$ LLM generates $Q_t$ and aspects simultaneously
14:
15:         **// Compute EVPI & Cost for Each Question**
16:         **for** each $q \in Q_t$ **do**
17:             $\text{EVPI}(q, \mathcal{B}(t)) = \mathbb{E}_{r \sim P(r|q, \mathcal{B}(t))} [\max_{c_i \in \mathcal{C}_t} \pi_i(t|q, r)] - \max_{c_i \in \mathcal{C}_t} \pi_i(t)$
18:             $\text{Cost}(q, t) = \lambda \sum_{a \in \mathcal{A}(q)} n_a(t)$             $\triangleright$ Redundancy penalty
19:             $\text{Score}(q) = \text{EVPI}(q, \mathcal{B}(t)) - \text{Cost}(q, t)$
20:         **end for**
21:
22:         **// Check Stopping Criterion**
23:         **if** $\max_{q \in Q_t} \text{Score}(q) < \alpha \cdot \max_{c_i \in \mathcal{C}_t} \pi_i(t)$ **then**
24:             **// Act: Execute Best Tool Call**
25:             $c^*(t) \leftarrow \arg\max_{c_i \in \mathcal{C}_t} \pi_i(t)$
26:             Execute $c^*(t)$ and **return** result
27:         **else**
28:             **// Act: Query User**
29:             $q^* \leftarrow \arg\max_{q \in Q_t} \text{Score}(q)$
30:             Query user with $q^*$ and receive response $o_{t+1}$
31:             $\boldsymbol{\pi}(t+1) \leftarrow \mathcal{O}b(\boldsymbol{\pi}(t), o_{t+1})$        $\triangleright$ Update beliefs via domain constraints
32:             $\mathcal{O}_{t+1} \leftarrow \mathcal{O}_t \cup \{o_{t+1}\}$
33:             **for** each $a \in \mathcal{A}(q^*)$ **do**
34:                 $n_a(t+1) \leftarrow n_a(t) + 1$             $\triangleright$ Update aspect history
35:             **end for**
36:         **end if**
37:     **else**
38:         **// Act: Execute Best Tool Call (Low Uncertainty)**
39:         $c^*(t) \leftarrow \arg\max_{c_i \in \mathcal{C}_t} \pi_i(t)$
40:         Execute $c^*(t)$ and **return** result
41:     **end if**
42: **end for**

---

criterion (line 24). If the maximum net information gain is insufficient, it executes the best candidate; otherwise, it poses the highest-scoring question, updates beliefs via domain constraint propagation (line 32), and increments aspect history (lines 34-36). When uncertainty is low, the agent executes the best candidate immediately (lines 41-43).

**Domain Constraint Propagation.** The belief update function $\mathcal{O}b$ (line 32) implements the constraint extraction function that maps natural language responses to parameter domain refinements: $\mathcal{D}_{i,j}(t+1) = \mathcal{D}_{i,j}(t) \cap C(r)$. This function handles:

- **Explicit constraints:** Direct specifications like "departure date is March 15th"

- **Schema dependencies:** Cross-parameter constraints where one parameter's value restricts available options for another parameter

- **Negative constraints:** Exclusions like "not business class" $\rightarrow$ class $\in \{$economy, premium$\}$

**Error Recovery Mechanism.** When the highest-confidence candidate fails at runtime, the system generates diagnostic questions using function $f_{\text{error}}(\cdot)$. This adaptive questioning strategy enables recovery from API failures, timeouts, and invalid parameter combinations that pass initial validation.

## A.3   PROMPTS

**Reasoning Prompt**   This prompt is used in the main reasoning phase of the ReAct agent to decide which tool to use next based on the current state of the conversation.

```
You are an AI assistant helping with a user request.
SYSTEM CONTEXT:
You have access to the following tool domain:
{plugin_descriptions}
Request: {request}
Previous observations:
{obs_text}
Available tools:
{tool_registry.get_tool_descriptions()}
Think step by step about what tool to use next. Consider the plugin
    context above to understand the capabilities available to you. If you
     have enough information to provide a final answer, use the
    final_answer tool.
Respond in JSON format:
{
"reasoning": "Your step-by-step thinking",
"tool_call": {
"tool_name": "name_of_tool",
"arguments": {
"arg1": "value1",
"arg2": "value2"
}
}
}
```

**Error Recovery Prompt**   Used when a tool execution fails to determine if the error can be resolved automatically.

```
You are helping fix a failed tool call.
Original Request: {request}
Tool Information:
{tool_info or f"Tool: {tool_name}"}
Error Details:
{error_result.message}
Based on the error and tool information, can you suggest how to fix this?
Respond in JSON format:
{
"can_fix": true/false,
"reasoning": "explanation of what went wrong and how to fix it",
"suggested_action": "retry_with_changes" or "different_tool" or "
    need_clarification",
"observation": "observation to add to context for next reasoning step"
}
If you cannot determine a fix from the available information, set can_fix
    to false.
```

**Question Generation Prompt**   Used to generate clarification questions when there is uncertainty about tool arguments.

```
You are an AI assistant that helps users by understanding their queries
    and executing tool calls.
{conversation_history}Original user query:
"{user_query}"
Based on the query, I've determined that the following tool calls are
    needed, but some arguments are uncertain:
Tool Calls:
{tool_calls}
Detailed Tool Documentation:
{tool_documentation}
Uncertain Arguments:
{uncertain_args}
Your task is to generate clarification questions that would help resolve
    the uncertainty about specific arguments.
Instructions:

Generate questions that are clear, specific, and directly address the
    uncertain arguments
Each question should target one or more specific arguments
Questions should be conversational and easy for a user to understand
For each question, specify which tool and argument(s) it aims to clarify.
Generate 5 diverse questions.
Keep in mind the the arguments you wish to clarify, their domains etc.

Return your response as a JSON object with the following structure:
{
"questions": [
{
"question": "A clear question to ask the user",
"target_args": [["tool_name", "arg_name"], ["tool_name", "other_arg_name
    "]]
}
// ... 5 total questions
]
}
Ensure that each question targets at least one uncertain argument.
```

## A.4 SENSITIVITY TO $\epsilon$

The parameter $\epsilon$ is used to quantify uncertainty for large domains, where the tool argument domain $|\mathcal{D}|$ is continuous or infinite. As long as the order of $\epsilon \ll 1/|\mathcal{D}_{\text{finite}}|$, the decisions are robust to the exact value of $\epsilon$, since scoring would switch unambiguously in favor of appropriate domains. However, very small values of $\epsilon$ may cause numerical instability, since it is exponentiated during computation.

We empirically validated the sensitivity to $\epsilon$ by retroactively checking for changes in question selection in our experiments from Section 8 on ClarifyBench (Ambiguous subset), using GPT-4o and Qwen2.5-14B-Instruct. We tested $\epsilon$ values: $\{10^{-6}, 10^{-5}, 10^{-4}, 10^{-3}, 10^{-2}, 0.05, 0.1, 0.2, 0.3, 0.4, 0.5, 0.6, 0.7, 0.8, 0.9\}$. As shown in Figure 7, when $\epsilon \geq 0.1$, the decisions diverge significantly, since domains are not effectively expressed as "infinite" when $\epsilon$ values are comparable to finite domain probabilities. However, for $\epsilon \leq 10^{-2}$, over 96–97% of decisions remain unchanged across all tested values, demonstrating robustness in the practical range.

# B REWARD MODELING WITH UNCERTAINTY

## B.1 DATASET PROCESSING

**Source Dataset:** Our enhanced dataset was constructed from the nvidia/When2Call dataset, from the "train_pref" data. This dataset contains preference-ranked examples for tool-calling tasks with human-annotated preferred responses for training reinforcement learning models.

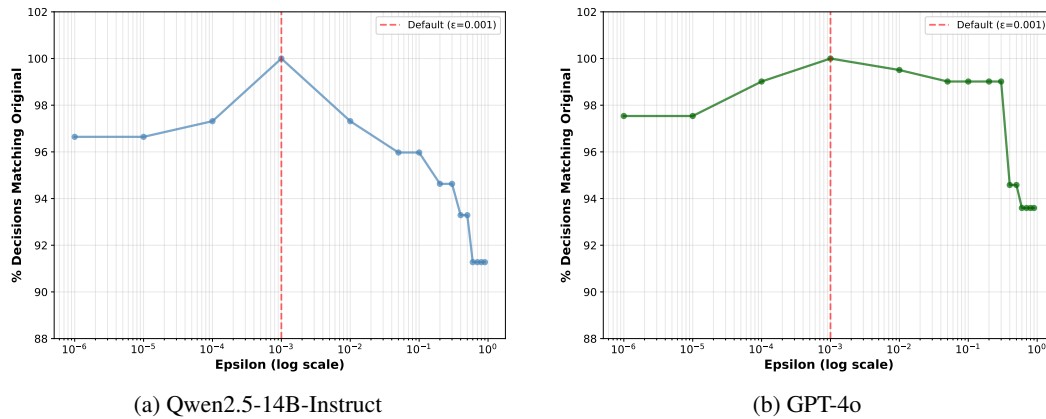

(a) Qwen2.5-14B-Instruct        (b) GPT-4o

Figure 7: Sensitivity analysis of $\epsilon$ on question selection decisions for the Ambiguous subset of ClarifyBench. The plots show the percentage of decisions that remain unchanged as $\epsilon$ varies across tested values, demonstrating robustness for $\epsilon \leq 10^{-2}$.

**Original Data Structure:** Each example in the source dataset contained:

- **Messages**:Conversation history with user and assistant exchanges in chat format
- **Tools**: Available tool definitions with JSON schema parameters and descriptions
- **Chosen responses**: Human-preferred responses for the given context
- **Preference annotations**: Quality ratings for different response options

**Response Classification:** Each example was processed to classify responses into four categories: `<TOOLCALL>`, `<ASK>`, `<REFUSE>`, and `<DIRECTLY>`. Classification used keyword-based heuristics:

- `<TOOLCALL>`: Presence of "<TOOLCALL>" tags or "toolcall" keywords
- `<ASK>`: Presence of question marks ("?") in content
- `<REFUSE>`: Presence of refusal keywords ("sorry", "unable", "impossible", etc.)
- `<DIRECTLY>`: Default classification for other responses. (None existed in the preferred set)

**Data Transformations:** Several preprocessing steps were applied to optimize the dataset for uncertainty-aware training:

1. **Domain Schema Injection**: Each example was augmented with parsed domain information for all available tools, stored as JSON strings in a `tool_domain_schemas` field for HuggingFace compatibility
2. **Message Format Preservation**: The chat format was maintained with modified system messages while preserving user/assistant alternation

### B.2 TOOL DOMAIN ANALYSIS

To enable uncertainty quantification, we performed comprehensive domain analysis of all available tools using Qwen-2.5-7B-Instruct as the primary analysis model. Each tool's arguments were analyzed to determine:

- **Domain type**: finite, estimated_finite, numeric_range, string, boolean, list, or custom
- **Domain size**: exact count for finite domains, estimates for larger domains, or infinite for unbounded domains
- **Domain values**: complete enumeration for small domains, representative examples for larger domains, or range bounds for numeric domains
- **Data dependency**: whether argument values depend on external data sources or user context

The analysis prompt instructed the model to classify arguments according to strict validation rules:

- Finite domains ($\leq 20$ values): complete value enumeration with domain_size = len(domain_values)
- Estimated finite domains: 5-10 representative examples with domain_size $>>$ len(examples)
- Numeric ranges: [min, max] bounds with appropriate size calculation
- Boolean domains: domain_size = 2 with null values
- String/custom domains: infinite size with null values

## B.3 UNCERTAINTY-AWARE SYSTEM PROMPTS

Each training example was enhanced with a comprehensive system prompt that provided explicit instructions for uncertainty handling. The complete system prompt template was:

```
\texttt{You are a helpful agent. You will have access to tools to answer
    the query.\\
\\
UNCERTAINTY GUIDELINES:\\
- Use <UNK> for arguments you cannot determine from context, or cannot
    reasonably estimate. Don't overuse, you can assume defaults where
    needed.\\
- When asking questions, use the structured format with candidate tool
    calls\\
\\
You can perform following action types:\\
a) <TOOLCALL> Invoke a tool call as follows:\\
<TOOLCALL>\\
[\{"name": "tool\_name", "arguments": \{"argument\_name": "value", "
    uncertain\_argument": "<UNK>", ...\}\}]\\
</TOOLCALL>\\
\\
b) <ASK> Ask a question from the user if you need more information to
    execute a tool call </ASK>\\
\\
STRUCTURED QUESTION FORMAT (when asking for clarification):\\
<ASK>\\
<TOOLCALL>\\
// Think about what tool you would call given the request, and the
    current information. Because some information is missing, you want to
     ask a question.\\
[
\{\{ "name": "tool\_name", "arguments": \{"known\_arg": "value", "
    uncertain\_arg": "<UNK>"\}\}]\\
</TOOLCALL>\\
<question>\\
What is the specific value for uncertain\_arg?\\
</question>\\
</ASK>\\
\\
c) <REFUSE> Refuse, if your knowledge or available tools can't be used
    here </REFUSE>\\
d) <DIRECTLY> directly answer </DIRECTLY>\\
\\
Your response should be formatted like:\\
<reasoning>\\
Step-by-step thinking about certainty/uncertainty of each argument\\
</reasoning>\\
<answer>\\
<ACTION\_TYPE>\\
..content.. (Question/ToolCall/Refuse/DirectAnswer)\\
</ACTION\_TYPE>\\
</answer>}
```

## B.4 TRAINING CONFIGURATION

Training began from `unsloth/Qwen2.5-3B-Instruct` and `unsloth/Qwen2.5-7B-Instruct` checkpoints. LoRA (Low-Rank Adaptation) fine-tuning was applied with rank 64 adaptations targeting attention and MLP projection layers.

Model training was performed using Group Relative Policy Optimization, using Unsloth (Daniel Han & team, 2023) with parameter details in Table 5.

| Hyperparameter | Value |
|---|---|
| Learning Rate | 5e-6 |
| Per Device Batch Size | 1 (3B), 8 (logs) |
| Gradient Accumulation Steps | 1 |
| Max Sequence Length | 1024 |
| Training Epochs | 1 |
| Warmup Ratio | 0.1 |
| Weight Decay | 0.1 |
| Optimizer | AdamW 8-bit |
| Adam Beta1 | 0.9 |
| Adam Beta2 | 0.99 |
| LoRA Rank | 64 |
| LoRA Alpha | 64 |

Table 5: Training hyperparameters for uncertainty-aware tool calling model.

## B.5 REWARD SPECIFICATION

Our baseline GRPO reward function consists of multiple components that guide the model toward generating well-formed, accurate responses. The total reward for a generated completion is computed as the sum of three independent reward components:

$$r_{\text{total}} = r_{\text{fmt}} + r_{\text{tool}} + r_{\text{cls}} \tag{7}$$

where $r_{\text{fmt}}$ represents format compliance rewards, $r_{\text{tool}}$ represents tool call accuracy, and $r_{\text{cls}}$ represents action classification rewards.

**Format Compliance Rewards** ($r_{\text{fmt}}$). These components encourage proper XML formatting and total up to 1.5 points:

- **XML Count Reward**: Awards up to 0.5 points for proper newline structure, penalizing excessive trailing content.

- **Soft Format Reward**: Awards 0.5 points if the response contains `<reasoning>` and `<answer>` tags in the correct order (with flexible whitespace).

- **Strict Format Reward**: Awards 0.5 points only if the response exactly matches the format `<reasoning>\n...\n</reasoning>\n<answer>\n...\n</answer>\n`.

**Tool Call Accuracy Reward** ($r_{\text{tool}}$). Compares the predicted tool call against a ground truth reference:

$$r_{\text{tool}} = \begin{cases} 1.0 & \text{if tool name and arguments match exactly} \\ 0.75 & \text{if tool name matches but arguments differ} \\ 0.5 & \text{if both have no tool call OR wrong tool name} \\ 0.0 & \text{if one has a tool call and the other does not} \end{cases} \tag{8}$$

**Action Classification Reward** ($r_{\text{cls}}$). This reward is the primary component that differentiates between GRPO and Certainty weighted GRPO. This reward is computed based on the agent's chosen action $a_t$ at timestep $t$, which can be: TOOLCALL (execute a tool), ASK (request clarification), REFUSE (decline the request), or DIRECTLY (answer without tools).

The base classification reward is computed as:

$$r_{\text{cls}}(a_t) = \begin{cases} 2.0 & \text{if response starts with correct tag and contains } \geq 30 \text{ chars} \\ 1.5 & \text{if response starts with correct tag but insufficient content} \\ 0.0 & \text{otherwise} \end{cases} \quad (9)$$

**Certainty Weighting** For the baseline **GRPO**, the final classification reward is simply:

$$r_{\text{cls}}^{\text{GRPO}}(a_t) = r_{\text{cls}}(a_t) \quad (10)$$

For **Certainty weighted GRPO**, we introduce epistemic-state-aware weighting. Let $\pi_c(t)$ be the model's belief over candidate tool calls $c \in \mathcal{C}_t$. We define the certainty function:

$$\text{Cert}(a_t) = \begin{cases} \max_c \pi_c(t) & \text{if } a_t \text{ is a tool call} \\ 1 - \max_c \pi_c(t) & \text{if } a_t \text{ is a clarification question} \\ 1 & \text{otherwise} \end{cases} \quad (11)$$

The final classification reward is then:

$$r_{\text{cls}}^{\text{Certainty}}(a_t) = \text{Cert}(a_t) \cdot r_{\text{cls}}(a_t) \quad (12)$$

This formulation up-weights confident correct tool calls, penalizes low-certainty calls, and rewards clarification only when uncertainty is high—thus aligning the reward with the agent's own epistemic state.

In our implementation, we approximate $\pi_c(t)$ through explicit certainty computation over tool call arguments. For a tool call $c$ with arguments, the certainty is:

$$\pi_c(t) = \prod_{\text{arg} \in c.\text{arguments}} \pi_{\text{arg}} \quad (13)$$

where for each argument:

$$\pi_{\text{arg}} = \begin{cases} 1.0 & \text{if arg has a specified value} \\ \frac{1}{|\mathcal{D}_{\text{arg}}|} & \text{if arg is empty and domain size is finite} \\ \epsilon \approx 0.0001 & \text{if arg is empty and domain size is infinite} \end{cases} \quad (14)$$

Here, $\mathcal{D}_{\text{arg}}$ represents the domain size for that argument as specified in the tool schema. This approach ensures that tool calls with all arguments specified receive maximum certainty ($\pi_c(t) = 1.0$), while tool calls with missing arguments receive certainty inversely proportional to the domain sizes of unspecified parameters. For ASK actions, we compute certainty over the candidate tool call mentioned in the question, and use $1 - \pi_c(t)$ to reward asking when uncertainty is high.

## C BENCHMARK DETAILS

### C.1 TASK FORMALIZATION

We formally define the clarification task as a multi-turn interaction problem between a tool-equipped agent and a user simulator within a structured environment.

#### C.1.1 PROBLEM DEFINITION

Let $\mathcal{E}$ denote the environment containing a set of tools $\mathcal{F} = \{f_1, f_2, \ldots, f_m\}$, where each tool $f_j$ has a signature defining its parameters and return type. An agent $\mathcal{A}$ is equipped with access to $\mathcal{F}$ and must satisfy user requests through appropriate tool invocations.

A simulation scenario $\mathcal{S}$ is defined as a tuple:

$$\mathcal{S} = \langle \mathcal{R}, \mathcal{I}, \mathcal{G}, \mathcal{K} \rangle \quad (15)$$

where:

- $\mathcal{R} = \{r_0, r_1, \ldots, r_n\}$ is a sequence of user requests
- $\mathcal{I}$ represents the true user intention for each request
- $\mathcal{G} = \{g_0, g_1, \ldots, g_n\}$ is the ground truth tool call sequence
- $\mathcal{K}$ is the knowledge being accumulated and used (conversational context, tool descriptions)

Each request $r_i \in \mathcal{R}$ belongs to one of three categories:

- **Normal**: Requests with sufficient information for direct execution
- **Ambiguous**: Requests requiring clarification to resolve uncertainty
- **Infeasible**: Requests that cannot be fulfilled with available tools

### C.1.2 AGENT AND USER SIMULATOR

The agent $\mathcal{A}$ takes as input the current query $q$ and conversation history $\mathcal{C}$, and produces one of three response types:

$$\mathcal{A}(q, \mathcal{C}) \to \begin{cases} \Phi_{success} & \text{tool call(s) executed} \\ \Phi_{clarification} & \text{clarifying question posed} \\ \Phi_{failure} & \text{task declined or failed} \end{cases} \tag{16}$$

The user simulator $\mathcal{U}$ maintains access to the true intention $\mathcal{I}$ and background knowledge $\mathcal{K}$. Given a clarifying question from the agent, the simulator responds:

$$\mathcal{U}(question, \mathcal{S}) \to \{clarification \quad \text{if answerable from } \mathcal{K}, \mathcal{I} \tag{17}$$

### C.1.3 MULTI-TURN INTERACTION PROCESS

The interaction proceeds as a sequence of turns $\mathcal{T}_i$ for each request $r_i$, as formalized in Algorithm 2. At each turn $t$, the agent either executes tool calls, poses a clarifying question, or declines the request. The query state is enriched with each clarification response:

$$q_{current}^{(t+1)} = \text{Enrich}(r_i, clarification^{(t)}) \tag{18}$$

To prevent infinite loops, we impose a maximum clarification threshold $\tau_{max}$ per request. The simulation maintains a conversation history $\mathcal{C}$ that accumulates all interaction turns across multiple requests, enabling the agent to leverage context from previous requests when handling subsequent ones.

## C.2 PROMPTS

### C.2.1 DATASET AUGMENTATION PROMPTS

The following prompt was used to augment user queries i.e. convert tool calls to corresponding user requests.

```
Original query: "{original_query}"

Tool call that should result from this query:
Tool: {tool_call["tool_name"]}
Parameters: {tool_call["parameters"]}

Update the query to naturally lead to these exact parameters.
The updated query should:
1. Be realistic and maintain the user's intent
2. Naturally incorporate the corrupted parameter value
3. Sound like something a real user would ask

Only return the updated query text, nothing else.
```

**Algorithm 2** ClarifyBench Interaction Protocol

---

1: **procedure** EXECUTESIMULATION($\mathcal{S}$)        $\triangleright$ $\mathcal{S}$ represents the simulation scenario
2:     Initialize agent $\mathcal{A}$, environment $\mathcal{E}$, user model $\mathcal{U}$
3:     $\mathcal{R} \leftarrow \{r_0, r_1, \ldots, r_n\}$        $\triangleright$ Request sequence
4:     $\mathcal{C} \leftarrow \emptyset$        $\triangleright$ Conversation history
5:     **for** each request $r_i \in \mathcal{R}$ **do**
6:        $\mathcal{T}_i \leftarrow \emptyset$        $\triangleright$ Turn sequence for request $i$
7:        $q_{current} \leftarrow r_i$        $\triangleright$ Current query state
8:        $clarification\_count \leftarrow 0$
9:        **while** $clarification\_count < \tau_{max}$ **and not** terminated **do**
10:           $response \leftarrow \mathcal{A}(q_{current}, \mathcal{C})$
11:           **if** $response \in \Phi_{success}$ **then**        $\triangleright$ Successful completion
12:              Record completion in $\mathcal{T}_i$
13:              **break**
14:           **else if** $response \in \Phi_{clarification}$ **then**        $\triangleright$ Needs clarification
15:              $clarification \leftarrow \mathcal{U}(response.question, \mathcal{S})$
16:              **if** $clarification = \bot$ **then**        $\triangleright$ User cannot provide clarification
17:                 Record incomplete in $\mathcal{T}_i$
18:                 **break**
19:              **end if**
20:              $q_{current} \leftarrow Enrich(r_i, clarification)$
21:              $clarification\_count \leftarrow clarification\_count + 1$
22:           **else**
23:              Record failure in $\mathcal{T}_i$
24:              **break**
25:           **end if**
26:        **end while**
27:        $\mathcal{C} \leftarrow \mathcal{C} \cup \mathcal{T}_i$
28:     **end for**
29:     **return** $\mathcal{C}$
30: **end procedure**

### C.2.2 USER SIMULATOR PROMPTS

The simulator takes a language model provider, ground truth data, and user intent as inputs. It maintains the conversation state and ensures responses are consistent with the user's information. The core of the simulation lies in two prompt templates that instruct a language model to act as a user:

```
You are simulating a user who is interacting with an AI assistant.
Original query: "{self.original_query}"
User's intent for the CURRENT request: {self.user_intent}
Information needed for the CURRENT request (do not reveal future
    intentions):
{current_turn_ground_truth}
Additional context:
{self.context}
The AI assistant has asked the following specific question:
"{question}"
Generate a realistic user response to this SPECIFIC question. The
    response should:

Be natural and conversational
ONLY provide information that directly answers the specific question
    asked
NOT mention any future requests or intentions the user might have
ONLY focus on the current task, not on future tasks
Be concise and to the point

IMPORTANT: Never reveal future intentions. Respond ONLY to the specific
    question asked.
NEVER BREAK CHARACTER. DO NOT THINK OUT LOUD. Respond directly as the
    user would:
```

This template ensures the simulator provides natural, conversational responses that only address the specific question without revealing future intentions. For generating follow-up requests, the simulator uses this template:

```
You are simulating a user who is interacting with an AI assistant.
Original query: "{self.original_query}"
User's intent: {self.user_intent}
Previous conversation:
{formatted_history}
Based on the conversation so far and the user's intent, decide if the
    user would have a follow-up request.
Consider:

Has everything the user wanted been accomplished?
Is there a logical next step the user might want to take?
Has the agent clearly indicated that they've completed all necessary
    tasks?

If you believe the user would have a follow-up request, provide it in a
    natural, conversational way.
If you believe the conversation is complete, respond with "
    CONVERSATION_COMPLETE".
NEVER BREAK CHARACTER, DO NOT THINK!
Decision:
```

This template helps the simulator determine whether to generate a follow-up request based on the conversation context and predefined potential follow-ups. The User Simulator isolates ground truth information for each conversation turn, ensuring only relevant information is revealed at appropriate times. It tracks the original query, user intent, ground truth for tool calls, completed tool calls, potential follow-up queries, and the current conversation turn. By providing consistent, realistic user responses, the simulator allows for reproducible evaluation of clarification strategies across multiple scenarios.

## C.3 BENCHMARK DOMAIN AREAS

This appendix describes the key characteristics of each API domain used in our experiments, detailing their initialization parameters, state management, and tool specifications.

**Gorilla File System Plugin (GFS).** The Gorilla File System API simulates a UNIX-like file system with a hierarchical directory structure. It maintains state through:

- Directory structure with nested files and subdirectories
- Current working directory pointer
- Each file contains content as strings

The plugin provides 18 tools implementing common file system operations such as navigation, file creation, modification, and content manipulation. Each tool supports parameters relevant to file system operations, such as file names, directory paths, and content strings. Table 10 provides detailed information about these tools and their parameter domains.

The GFS plugin's domains depend heavily on the current state of the file system. Domain updates revolve primarily around available files and directories in the current working directory, as outlined in Table 11.

**Document Processing.** The Document API simulates operations for PDF document manipulation. Its state consists of:

- Number of pages in the current document
- PDF filename metadata
- Operation-specific context for page-based operations

The plugin provides 18 document manipulation tools including conversion, annotation, redaction, and page manipulation functions. Parameters include page numbers, text content, formatting options, and file paths. Table 7 details the tools and their parameter domains.

Domain updates in the Document Plugin focus on page numbers and ranges, adapting dynamically to changes in document length when pages are added or deleted, as shown in Table 11.

**Vehicle Control.** The Vehicle Control API simulates an automotive control system with:

- Engine state (running or stopped)
- Door lock status for each door
- Fuel level (ranging from 0 to 50 gallons)
- Battery voltage
- Climate control settings
- Brake systems (pedal position and parking brake)
- Lighting systems
- Navigation state

This plugin implements 24 vehicle control tools that manipulate different aspects of the vehicle, including engine operations, door management, climate control, lighting, braking systems, and navigation. Table 9 details the specific tools and their parameter domains.

Vehicle Control domain updates primarily concern contextual constraints such as brake pedal position for engine start, door states, and fuel level requirements, as referenced in Table 11.

**Travel.** The Travel API simulates a travel booking and management system with:

- Credit card registry and balances
- Flight booking records
- User information (first name, last name)
- Budget limits
- Available routes with pricing data

The plugin provides 15 tools for travel-related operations, including flight bookings, credit card management, budget settings, and travel information queries. Table **??** details these tools and their parameter domains.

Domain updates in the Travel Plugin focus on available credit cards, booking IDs, and airport codes for valid routes, as detailed in Table 11.

**Trading Bot.** The Trading Bot simulates a stock trading platform with:

- Account information and balance
- Order records (pending, completed, cancelled)
- Stock data with prices and metrics
- Watchlist of stocks
- Transaction history
- Market status (open/closed)

This plugin provides 19 trading tools for account management, order placement, stock information retrieval, and market analysis. Table 8 lists the specific tools and their parameter domains.

Trading Plugin domain updates primarily involve available stocks, watchlist items, and order IDs, adapting to user actions like placing orders or modifying watchlists, as referenced in Table 11.

All plugins follow a consistent pattern for state initialization through configuration objects, domain updates based on state changes, and parameter validation. The dynamic nature of these domains presents particular challenges for language model interactions, as valid parameter values continuously evolve during conversations based on system state changes.

### C.4 HUMAN ANNOTATION

We employed two graduate student annotators, aged 22-25. The annotators were proficient in English, and have proficiency in Python (relevant to test tool calls). The annotators were fairly compensated at the standard Graduate Assistant hourly rate, following their respective graduate school policies. Fig 8 shows a summary of the annotator guidelines. Two annotators assign a 5-point Likert score to every candidate query, and the final selected query for a sample is the one that receives the highest score. Inter-annotator agreement for the highest-scoring selections is given by Cohen's $\kappa = 0.76$.

### C.5 TOOL CALL CORRUPTION HEURISTICS

We handcrafted rues to corrupt validated tool calls in the ground truth data, to construct ClarifyBench-Infeasible.

**GorillaFileSystem** For the file system API, we implemented four primary corruption strategies:

- *Invalid File Name Corruption* targeting functions like `mkdir`, `touch`, and `cat` by inserting forbidden characters (e.g., |, /, \, ?);
- *Path Traversal Corruption* for `cd`, `mv`, `cp`, and `find` operations by inserting relative paths (`../`) or absolute paths (`/root/`);

| Tool Name | Argument | Description | Domain Type | Domain Values | Data Dep. | Required |
|---|---|---|---|---|---|---|
| get_budget_fiscal_year | lastModifiedAfter | Date filter for fiscal years | string | Any date string | N | N |
| | includeRemoved | Include removed fiscal years | string | Any string | N | N |
| register_credit_card | card_number | Credit card number | string | Any card number | N | Y |
| | expiration_date | Card expiration (MM/YYYY) | string | MM/YYYY format | N | Y |
| | cardholder_name | Name on card | string | Any name string | N | Y |
| | card_verification_number | CVV code | numeric_range | [100, 999] | N | Y |
| get_flight_cost | travel_from | Departure airport code | string* | 3-letter codes | Y | Y |
| | travel_to | Arrival airport code | string* | 3-letter codes | Y | Y |
| | travel_date | Travel date | string | YYYY-MM-DD | N | Y |
| | travel_class | Seat class | finite | [economy, business, first] | N | Y |
| get_credit_card_balance | card_id | Credit card identifier | string* | Card ID list | Y | Y |
| book_flight | card_id | Payment card ID | string* | Card ID list | Y | Y |
| | travel_date | Travel date | string | YYYY-MM-DD | N | Y |
| | travel_from | Departure airport | string* | Airport codes | Y | Y |
| | travel_to | Arrival airport | string* | Airport codes | Y | Y |
| | travel_class | Seat class | finite | [economy, business, first] | N | Y |
| | travel_cost | Flight cost | numeric_range | [0, 10000] | N | Y |
| retrieve_invoice | booking_id | Booking identifier | string* | Booking ID list | Y | N |
| | insurance_id | Insurance identifier | string* | Insurance ID list | Y | N |
| list_all_airports | | | | *No arguments* | | |
| cancel_booking | booking_id | Booking to cancel | string* | Booking ID list | Y | Y |
| compute_exchange_rate | base_currency | Source currency | finite | [USD, RMB, EUR, JPY, GBP, CAD, AUD, INR, RUB, BRL, MXN] | N | Y |
| | target_currency | Target currency | finite | [USD, RMB, EUR, JPY, GBP, CAD, AUD, INR, RUB, BRL, MXN] | N | Y |
| | value | Amount to convert | numeric_range | [0, 1000000] | N | Y |
| verify_traveler_information | first_name | Traveler's first name | string | Any name | N | Y |
| | last_name | Traveler's last name | string | Any name | N | Y |
| | date_of_birth | Birth date | string | YYYY-MM-DD | N | Y |
| | passport_number | Passport number | string | Any passport ID | N | Y |
| set_budget_limit | budget_limit | Budget limit in USD | numeric_range | [0, 10000] | N | Y |
| get_nearest_airport_by_city | location | City name | finite | [Rivermist, Stonebrook, ...] | N | Y |
| purchase_insurance | insurance_type | Type of insurance | finite | [basic, premium, deluxe] | N | Y |
| | booking_id | Booking identifier | string* | Booking ID list | Y | Y |
| | insurance_cost | Insurance cost | numeric_range | [0, 1000] | N | Y |
| | card_id | Payment card ID | string* | Card ID list | Y | Y |
| contact_customer_support | booking_id | Booking reference | string* | Booking ID list | Y | Y |
| | message | Support message | string | Any message text | N | Y |
| get_all_credit_cards | | | | *No arguments* | | |

Table 6: Travel Plugin API: Complete Tool and Argument Specification with Domain Dependencies (without Importance column)

- *Non-existent Files Corruption* for file operation functions by generating random names or modifying existing names;
- *Duplicate Creation Corruption* for mkdir and touch operations by using existing file/directory names.

**DocumentPlugin** For the document manipulation API, we implemented three corruption strategies:

- *Invalid Page Range Corruption* for functions like add_comment and delete_page by setting zero/negative values or exceeding total pages;
- *Invalid Formats Corruption* for convert operations by using unsupported formats or partial strings;
- *Out of Range Values Corruption* for parameters like font_size and transparency by exceeding min/max bounds or using negative values.

**VehicleControlAPI** For the vehicle control API, we focused on two corruption categories:

- *Invalid Ranges Corruption* for functions like fillFuelTank and adjustClimateControl by exceeding capacity or using negative values;
- *Invalid Enums Corruption* for operations like startEngine and setHeadlights by supplying wrong enum values or case mismatches.

**TravelAPI** For the travel booking API, we implemented three corruption strategies:

- *Financial Constraints Corruption* for functions like book_flight by exceeding available balance or using negative values;
- *Invalid Routes Corruption* for route parameters by using non-existent airport codes or identical from/to locations;

| Tool Name | Argument | Description | Domain Type | Domain Values | Data Dep. | Required |
|---|---|---|---|---|---|---|
| duplicate | output_filename | Name of duplicate file | string | Any filename | N | Y |
| rename | output_filename | New filename | string | Any filename | N | Y |
| search | object_name | Search term/object | string | Any search term | N | Y |
| count_pages | | *No arguments* | | | | |
| compress_file | output_filename | Compressed output name | string | Any filename | N | N |
| convert | format | Target format | finite | [pptx, doc, png, jpeg, tiff] | N | Y |
| | output_filename | Output filename | string | Any filename | N | Y |
| | zip | Zip output files | boolean | [true, false] | N | N |
| add_comment | page_num | Page number | numeric_range* | [1, num_pages] | Y | Y |
| | coordinates | Comment position [x,y] | list | [x, y] coordinates | N | Y |
| | font_size | Font size (points) | numeric_range | [8, 72] | N | Y |
| redact_page_range | start | Start page (inclusive) | numeric_range* | [1, num_pages] | Y | Y |
| | end | End page (inclusive) | numeric_range* | [1, num_pages] | Y | Y |
| redact_text | start | Start page | numeric_range* | [1, num_pages] | Y | Y |
| | end | End page | numeric_range* | [1, num_pages] | Y | Y |
| | object_name | Text to redact (list) | list | List of text strings | N | Y |
| | overwrite | Overwrite original | boolean | [true, false] | N | Y |
| | output_pathname | Output filename | string | Any filename | N | N |
| highlight_text | start | Start page | numeric_range* | [1, num_pages] | Y | Y |
| | end | End page | numeric_range* | [1, num_pages] | Y | Y |
| | object_name | Text to highlight (list) | list | List of text strings | N | Y |
| | overwrite | Overwrite original | boolean | [true, false] | N | Y |
| | output_pathname | Output filename | string | Any filename | N | N |
| underline_text | start | Start page | numeric_range* | [1, num_pages] | Y | Y |
| | end | End page | numeric_range* | [1, num_pages] | Y | Y |
| | object_name | Text to underline (list) | list | List of text strings | N | Y |
| | overwrite | Overwrite original | boolean | [true, false] | N | Y |
| | output_pathname | Output filename | string | Any filename | N | N |
| extract_pages | start | Start page | numeric_range* | [1, num_pages] | Y | Y |
| | end | End page | numeric_range* | [1, num_pages] | Y | Y |
| | overwrite | Overwrite original | boolean | [true, false] | N | Y |
| | output_pathname | Output filename | string | Any filename | N | N |
| delete_page | page_num | Page to delete | numeric_range* | [1, num_pages] | Y | Y |
| | overwrite | Overwrite original | boolean | [true, false] | N | Y |
| | output_pathname | Output filename | string | Any filename | N | N |
| delete_page_range | start | Start page | numeric_range* | [1, num_pages] | Y | Y |
| | end | End page | numeric_range* | [1, num_pages] | Y | Y |
| | overwrite | Overwrite original | boolean | [true, false] | N | Y |
| | output_pathname | Output filename | string | Any filename | N | N |
| add_signature | page_num | Page for signature | numeric_range* | [1, num_pages] | Y | Y |
| | position | Signature position | finite | [top-left, top-middle, ...] | N | Y |
| | overwrite | Overwrite original | boolean | [true, false] | N | Y |
| | output_pathname | Output filename | string | Any filename | N | N |
| add_page_with_text | text_content | Page text content | string | Any text content | N | Y |
| | font_size | Text font size | numeric_range | [8, 72] | N | Y |
| | page_num | Insert position | numeric_range* | [1, num_pages+1] | Y | Y |
| add_watermark | watermark_text | Watermark text | string | Any text | N | Y |
| | transparency | Transparency level | numeric_range | [0.0, 1.0] | N | Y |
| add_password | password | PDF password | string | Any password string | N | Y |

Table 7: Document Plugin API: Complete Tool and Argument Specification with Domain Dependencies

| Tool Name | Argument | Description | Domain Type | Domain Values | Data Dep. | Required |
|---|---|---|---|---|---|---|
| get_current_time | | | | *No arguments* | | |
| update_market_status | current_time_str | Time in HH:MM AM/PM | string | HH:MM AM/PM format | N | Y |
| get_symbol_by_name | name | Company name | string | Any company name | N | Y |
| get_stock_info | symbol | Stock symbol | string* | Available stock symbols | Y | Y |
| get_order_details | order_id | Order identifier | numeric_range* | Existing order IDs | Y | Y |
| cancel_order | order_id | Order to cancel | numeric_range* | Existing order IDs | Y | Y |
| place_order | order_type | Buy or Sell | finite | [Buy, Sell] | N | Y |
| | symbol | Stock symbol | string* | Available stocks | Y | Y |
| | price | Price per share | numeric_range | [0.01, 10000.0] | N | Y |
| | amount | Number of shares | numeric_range | [1, 10000] | N | Y |
| make_transaction | xact_type | Transaction type | finite | [deposit, withdrawal] | N | Y |
| | amount | Transaction amount | numeric_range | [0.01, 1000000.0] | N | Y |
| get_account_info | | | | *No arguments* | | |
| fund_account | amount | Funding amount | numeric_range | [0.01, 1000000.0] | N | Y |
| remove_stock_from_watchlist | symbol | Stock to remove | string* | Watchlist stocks | Y | Y |
| get_watchlist | | | | *No arguments* | | |
| get_order_history | | | | *No arguments* | | |
| get_transaction_history | start_date | Start date filter | string | YYYY-MM-DD format | N | N |
| | end_date | End date filter | string | YYYY-MM-DD format | N | N |
| update_stock_price | symbol | Stock symbol | string* | Available stocks | Y | Y |
| | new_price | New stock price | numeric_range | [0.01, 10000.0] | N | Y |
| get_available_stocks | sector | Market sector | finite | [Technology, Automobile, Healthcare, Finance, Energy] | N | Y |
| filter_stocks_by_price | stocks | Stock list to filter | list | List of stock symbols | N | Y |
| | min_price | Minimum price | numeric_range | [0.01, 10000.0] | N | Y |
| | max_price | Maximum price | numeric_range | [0.01, 10000.0] | N | Y |
| add_to_watchlist | stock | Stock to add | string* | Available stocks | Y | Y |
| notify_price_change | stocks | Stocks to monitor | list | List of stock symbols | N | Y |
| | threshold | Change threshold (%) | numeric_range | [0.01, 100.0] | N | Y |

Table 8: Trading Plugin API: Complete Tool and Argument Specification with Domain Dependencies

| Tool Name | Argument | Description | Domain Type | Domain Values | Data Dep. | Required |
|---|---|---|---|---|---|---|
| startEngine | ignitionMode | Engine ignition mode | finite | [START, STOP] | N | Y |
| fillFuelTank | fuelAmount | Fuel to add (gallons) | numeric_range* | [0, 50-current_fuel] | Y | Y |
| lockDoors | unlock | Lock or unlock | boolean | [true, false] | N | Y |
| | door | Doors to operate | list* | [driver, passenger, rear_left, rear_right] | Y | Y |
| adjustClimateControl | temperature | Target temperature | numeric_range | [-10, 50] | N | Y |
| | unit | Temperature unit | finite | [celsius, fahrenheit] | N | N |
| | fanSpeed | Fan speed (0-100) | numeric_range | [0, 100] | N | N |
| | mode | Climate mode | finite | [auto, cool, heat, defrost] | N | N |
| get_outside_temperature_from_google | | | | *No arguments* | | |
| get_outside_temperature_from_weather_com | | | | *No arguments* | | |
| setHeadlights | mode | Headlight mode | finite | [on, off, auto] | N | Y |
| displayCarStatus | option | Status display option | finite | [fuel, battery, doors, climate, headlights, parkingBrake, brakePedal, engine] | N | Y |
| activateParkingBrake | mode | Brake mode | finite | [engage, release] | N | Y |
| pressBrakePedal | pedalPosition | Pedal position (0-1) | numeric_range | [0, 1] | N | Y |
| releaseBrakePedal | | | | *No arguments* | | |
| setCruiseControl | speed | Cruise speed (mph) | finite* | [0, 5, 10, ..., 120] | Y | Y |
| | activate | Activate cruise | boolean* | [true, false] | Y | Y |
| | distanceToNextVehicle | Following distance (m) | numeric_range | [0, 1000] | N | Y |
| get_current_speed | | | | *No arguments* | | |
| display_log | messages | Log messages | list | List of strings | N | Y |
| estimate_drive_feasibility_by_mileage | distance | Distance in miles | numeric_range | [0, 10000] | N | Y |
| liter_to_gallon | liter | Liters to convert | numeric_range | [0, 1000] | N | Y |
| gallon_to_liter | gallon | Gallons to convert | numeric_range | [0, 1000] | N | Y |
| estimate_distance | cityA | First city zipcode | finite | [83214, 74532, 56108, ...] | N | Y |
| | cityB | Second city zipcode | finite | [83214, 74532, 56108, ...] | N | Y |
| get_zipcode_based_on_city | city | City name | finite | [Rivermist, Stonebrook, ...] | N | Y |
| set_navigation | destination | Destination address | string | Street, city, state format | N | Y |
| check_tire_pressure | | | | *No arguments* | | |
| find_nearest_tire_shop | | | | *No arguments* | | |

Table 9: Vehicle Control Plugin API: Complete Tool and Argument Specification with Domain Dependencies

| Tool Name | Argument | Description | Domain Type | Domain Values | Data Dep. | Required |
|-----------|----------|-------------|-------------|---------------|-----------|----------|
| pwd | | *No arguments* | | | | |
| ls | a | Show hidden files | boolean | [true, false] | N | N |
| cd | folder | Directory to change to | string* | Available directories + [.., /] | Y | Y |
| mkdir | dir_name | New directory name | string | Any valid directory name | N | Y |
| touch | file_name | New file name | string | Any valid filename | N | Y |
| echo | content | Text content | string | Any text string | N | Y |
| | file_name | Output file (optional) | string | Any filename | N | N |
| cat | file_name | File to display | string* | Available files | Y | Y |
| find | path | Search starting point | string | Any path | N | N |
| | name | Search pattern | string | Any search pattern | N | N |
| wc | file_name | File to count | string* | Available files | Y | Y |
| | mode | Count mode | finite | [l, w, c] | N | N |
| sort | file_name | File to sort | string* | Available files | Y | Y |
| grep | file_name | File to search | string* | Available files | Y | Y |
| | pattern | Search pattern | string | Any text pattern | N | Y |
| du | human_readable | Human readable format | boolean | [true, false] | N | N |
| tail | file_name | File to display | string* | Available files | Y | Y |
| | lines | Number of lines | numeric_range | [1, 100] | N | N |
| diff | file_name1 | First file | string* | Available files | Y | Y |
| | file_name2 | Second file | string* | Available files | Y | Y |
| mv | source | Source file/directory | string* | Available items | Y | Y |
| | destination | Destination name | string* | Available items + new names | Y | Y |
| rm | file_name | File/directory to remove | string* | Available items | Y | Y |
| rmdir | dir_name | Directory to remove | string* | Available directories | Y | Y |
| cp | source | Source file/directory | string* | Available items | Y | Y |
| | destination | Destination name | string* | Available items + new names | Y | Y |

Table 10: File System Plugin API: Complete Tool and Argument Specification with Domain Dependencies

| Plugin | Update Trigger | Dynamic Domain Updates | Affected Operations |
|--------|----------------|------------------------|---------------------|
| **Travel** | | | |
| | Credit card registration | Card IDs → available payment methods | book_flight, get_credit_card_balance, purchase_insurance |
| | Flight booking | Booking IDs → cancellable/retrievable bookings | cancel_booking, retrieve_invoice, contact_customer_support |
| | Budget setting | Budget limits → financial constraints | All cost-related operations |
| | Route updates | Airport codes → valid travel routes | get_flight_cost, book_flight |
| **Document** | | | |
| | Page operations | Page count → valid page numbers | All page-specific operations |
| | Document loading | Total pages → range constraints | add_comment, delete_page, etc. |
| | Cache invalidation | State changes → domain refresh | Page-changing operations |
| **Trading** | | | |
| | Order placement | Order IDs → manageable orders | get_order_details, cancel_order |
| | Stock updates | Available stocks → tradeable symbols | place_order, get_stock_info |
| | Watchlist changes | Watchlist → removable stocks | remove_stock_from_watchlist |
| **Vehicle** | | | |
| | Fuel level changes | Current fuel → addable amount | fillFuelTank |
| | Door state changes | Door status → operable doors | lockDoors |
| | Engine state | Running/stopped → cruise control availability | setCruiseControl |
| **File System** | | | |
| | Directory navigation | Current contents → available items | cd, cat, mv, cp, rm |
| | File operations | File list → operable files | File-specific operations |
| | Directory changes | Directory list → navigable paths | cd, rmdir |
| | State synchronization | FS changes → domain cache invalidation | All state-changing operations |

Table 11: Dynamic Domain Update Rules and Triggers Across Plugin System

# Human Annotation Guidelines

**Objective:**
Annotators must evaluate five LLM-generated queries per sample. Each query is scored on three dimensions: (A) Naturalness of language, (B) Faithfulness to the expected tool call, and (C) Executability/Validity. Additionally, annotators must check for removal of Personally Identifiable Information (PII), assess tool call feasibility, and select one optimal query per sample.

**Evaluation Rubric**

| Criterion | Score 5 | Score 4 | Score 3 | Score 2 | Score 1 |
|---|---|---|---|---|---|
| **A. Naturalness** | Fully fluent, natural, human-like | Minor awkwardness or stiffness | Understandable but robotic | Clearly awkward or difficult to read | Unintelligible or nonsensical |
| **B. Faithfulness** | Perfect match to expected tool call; all required arguments present | Mostly aligned; minor phrasing or parameter issues | Some omissions or hallucinations; core logic intact | Major deviations from expected tool behavior | Entirely incorrect or misleading tool structure |
| **C. Executability** | Fully executable; properly structured and valid | Executes with minor issues or missing defaults | Partially executable with moderate corrections needed | Major issues preventing execution | Unexecutable or contradicts tool logic/API |

**Required Checks**

- **PII Removal:** Ensure no personal identifiers (names, emails, phone numbers, IDs) are present Flag these queries for further processing.

- **Tool Call Validation:** If feasible, simulate or run tool calls to confirm validity and argument correctness.

- **Error Identification:** Mark and annotate any queries with logical inconsistencies, invalid parameters, or unsupported constraints.

Figure 8: Summary of instructions given to human annotators.

- *Non-existent Booking Corruption* for functions like `cancel_booking` by generating random non-existent IDs.

**TradingBot**  For the stock trading API, we implemented three corruption strategies:

- *Invalid Symbols Corruption* for functions like `get_stock_info` by using non-existent symbols or malformed formats;
- *Financial Validation Corruption* for `place_order` and related functions by using negative values or amounts exceeding account balance;
- *Order State Conflicts Corruption* for `cancel_order` operations by referencing completed orders or using malformed order IDs.

