# OpenReview forum: "Structured Uncertainty guided Clarification for LLM Agents"
_ICLR.cc/2026/Conference — ICLR 2026 Conference Withdrawn Submission_

### Official Review · Reviewer_e7rp · 2025-10-14

**Soundness:** 2
**Presentation:** 1
**Contribution:** 3
**Rating:** 2
**Confidence:** 4

**Summary:**

The paper’s general topic are tool-using agents that have to ask clarification questions about ambiguous queries. It is split in three parts: 1) A handcrafted inference strategy to decide which clarification question to ask based on how complete the best possible function call would be, 2) An evaluation dataset with ambiguous scenarios, and 3) an RL-trained agent. The authors find that their two agents work better than their respective baselines.

**Strengths:**

1. The general topic is highly important, and the dataset is useful for future research
2. I appreciate that the authors always report spans of results and show confidence intervals and noise, even when broad.
3. Both proposed methods look promising, but it is hard to say how robust their performance is due to the limited evaluation (see below)

**Weaknesses:**

Weaknesses, in order of magnitude:
1. The paper seems to contain three parts that are only loosely connected by their general topic. For example, there is both the hand-engineered SAGE agent and then the GRPO trained agent in Section 6. They are similar in that they both use pi_i, but are never compared to one another or executed on the same benchmark dataset.
2. I think the paper would greatly benefit from being split into multiple papers, because:
    1. As a dataset paper, the dataset construction would need to be more critically assessed, and should have a larger size than n=716. The construction of the ClarifyBench dataset is also just very high-level and not detailed, neither in Section 4.2 (“we design handwritten rules based on common API errors to create tool calls that would generate failures, followed by a similar LLM-based query augmentation process”), nor in the appendix.
    2. As a paper that proposes a new method (SAGE-Agent), the new method is only evaluated on one dataset (the dataset introduced above), and there is a second method intoduced (GRPO training on When2Call, and evaluated on a different dataset), plus the methods failure modes and when it does / does not work are not analyzed.
3. The probabilities pi_i, upon which both the SAGE-Agent and the GRPO training rest, are heavily hand-crafted and do not take into account LLM’s token uncertainties.
4. The expectation over a maximum in the EVPI objective creates a heavy inference-time compute burden, especially in less restricted domains, because we have to search over all questions, take an expectation over all follow-up answers, and then search the maximum pi_i.
5. There are multiple readability issues. Normally I would note this as not influencing my score, but here it poses real difficulties to comprehend the paper. The paper could have a much greater impact if it benefitted from more time spent to make it readable.
    1. abbreviations are not introduced before usage (POMDP),
    2. the contributions section in the introduction is hard to understand due to the non-motivated and non-explained terms (Bayesian Value of Information Objective, expected value of perfect information),
    3. What are coverage rate, tool match rate, and parameter match rate (benchmark metrics in Table 3)
    4. observations_t in line 153 is undefined (and probably the same as obs_t)
    5. I suppose that line 117 is an indicator function?
    6. Cost(q) in line 177 is not defined until line 215
    7. There are Latex missing citations errors (line 181)
    8. Likewise, Algorithm 1 (called SAGE (final corrected version)) has multiple reference errors in the equations
    9. The reward function in line 177 is not used. A second reward is defined in line 400
    10. Citations are not in brackets and not hyperlinked,
    11. Section 4.2: Data Augmentation has multiple grammar errors
    12. vspace around figures is sometimes very small and sometimes very large (Figure 2 vs Figure 4),
6. I have doubts about the independence of the human annotators that ensure quality and naturalness of the dataset, given that it is “two graduate student annotators that were compensated following their respective graduate school policies” (Appendix B.2). To make the evaluation more rigid, I suggest to use independent raters (Mechanical turks and the likes), use multiple annotators per sample, and measure annotator disagreement.
7. It would be great to evaluate on more than one LLM (GPT-4o for SAGE-Agent and Qwen 2.5 3B/7B for the RL experiments). Especially in user simulation and interactive settings, different LLMs behave wildly differently.


Smaller weaknesses that do not influence my score and do not need to be rebuttled, but I suggest fixing them for the revised version:

* It would increase readability if you could hyperlink your references (e.g., use the cleveref package)
* The caption of Table 2 is not a full sentence
* Algorithm 1 has multiple fonts
* Spelling error in Table 3: LLm → LLM

## Justification for the overall score

I believe that this paper addresses an important gap in the current research field, clarification questions for tool-calling agents. However, the paper in its current form is not clear and reproducible, and strechted too thinly across three subtopics to discuss either of them in detail. It reads more like three workshop short-papers in the current form. I recommend to reject the paper at this point, but believe that the authors are working on something that is promising. I encourage the authors to disentangle their three subcomponents, and focus on one of them fully, in order to have the capacity to analyze and discuss it in detail.

**Questions:**

1. Could you explain the argument you make in lines 131-136? Why do other approaches _have to_ take this route to model ambiguity, and why is your approach not influenced by model (/epistemic) uncertainty? Your proposed uncertainties can easily break if the model makes a wrong prediction, this is assumed away if I understand correctly?
2. Can you discuss the difference between the Expected Value of Perfect Information and the more commonly used Expected Information Gain objective, and why the former corresponds to “optimal” (line 182) question selection in your opinion? Can you benchmark against Expected Information Gain?
3. By “aspect” a_i,j, you just mean the identifier (i, j) of some theta_i,j, is that correct?
4. Why is your reward self-calibrating? (line 405) How does this relate to calibration, is it a proper scoring rule?

---

> ### Author Response · Authors · 2025-11-27
>
> Thank you for your thoughtful and detailed review. We genuinely appreciate your recognition that our work "addresses an important gap in the current research field" and that "the general topic is highly important, and the dataset is useful for future research." Your acknowledgment that "both proposed methods look promising" and your detailed feedback have been invaluable in improving our manuscript.
>
> ---
>
> ## Addressing Major Weaknesses
>
> ### W1: Paper Structure and Cohesion
>
> We have updated the manuscript to better reflect how our contributions tie together. **The unifying contribution is structured uncertainty quantification for tool-calling disambiguation.** We demonstrate two complementary applications of this single framework:
>
> 1. **SAGE-Agent** (Sections 5): Uses structured uncertainty for inference-time question selection via EVPI
> 2. **Uncertainty-guided GRPO** (Section 6): Uses the same uncertainty measure (π_c) to provide training signals for learning when to ask versus act
>
> These contributions are fundamentally connected because both use the identical uncertainty formulation defined in Definition 3 and updated through Equations 1-2. The training objective in Section 6.2 directly implements Cert(a_t) using π_c(t) from Section 3.1. This demonstrates that structured uncertainty is not merely a hand-crafted heuristic, but a learnable principle that improves sample efficiency, as shown in Figure 6 where uncertainty-weighted GRPO achieves 65.2% accuracy compared to 36.5% for the baseline.
>
> We have restructured the manuscript to emphasize this unified framework. The Abstract (lines 21-30) now highlights "the versatility of this formulation through two applications." The Introduction contributions section (lines 78-89) explicitly connects inference and training applications through structured uncertainty.
>
>
> **ClarifyBench** (Section 4) enables evaluation of agentic scaffolds, by providing the first benchmark with dynamic user simulation for multi-turn disambiguation scenarios, which is essential for validating clarification strategies in realistic interactive settings.
>
> Regarding the lack of direct comparison between SAGE-Agent and GRPO models: the GRPO-trained models (Qwen 3B/7B) are optimized for the When2Call format and struggle with ClarifyBench's long-context, multi-turn simulations requiring different response structures. Conversely, SAGE-Agent is an inference-time method requiring no training. We now evaluate both on When2Call (Table 4, lines 486-499), showing they excel in their respective settings while sharing the same theoretical foundation.
>
>
> ### W2: Unified Contributions with Comprehensive Detail
>
> **We respectfully disagree that the paper should be split.** We appreciate your recognition of the novelty and believe showing these aspects together is essential to demonstrate that structured uncertainty is a general framework, not a collection of unrelated heuristics. We have taken your feedback seriously and significantly modified the manuscript to provide comprehensive detail for each contribution while maintaining their meaningful connection.
>
> #### Dataset Construction
>
> **Addressing the construction detail concerns:** You specifically noted that Section 4.2 was too high-level, mentioning only "handwritten rules based on common API errors" without sufficient detail. We have now provided comprehensive construction details at two levels:
>
> **Main paper (Section 4):** We modified this section to provide clearer motivation and overview. Section 4 begins by introducing the benchmark and comparing it against existing agent evaluation benchmarks to establish need (Table 1). Section 4.1 describes the high-level design including the three query types (explicit, ambiguous, infeasible) and multi-domain structure. Section 4.2 now describes data sources (DocPilot and BFCL-v3), provides clearer description of data augmentation strategies for each query type, and details the human validation process. We augmented this section by adding task formalization (lines 223-231) and discussing inter-annotator agreement (line 290). Table 2 provides statistical description across all five domains.

---

> > ### Author Response · Authors · 2025-11-27
> >
> > **Appendix C (9 pages of dataset details):** We reorganized the appendix to provide complete construction details enabling full reproduction:
> >
> > - **Section C.1 (lines 1080-1187):** Formal task definition including problem formalization (Equations 15-18), agent and user simulator specifications, and the multi-turn interaction protocol (Algorithm 2)
> >
> > - **Section C.1.4 (lines 1188-1241):** Complete LLM prompts for query augmentation showing exact instructions, obfuscation strategies for ambiguous queries, and user simulator prompts enabling dynamic interaction
> >
> > - **Sections C.2-C.3 (Tables 6-11):** Complete breakdown of all five domains including full API schemas for all 92 tools with parameter specifications, domain types, and constraints. Table 11 details domain dependency tracking showing how parameter domains update dynamically
> >
> > - **Section C.4 (Figure 8):** Complete human validation protocol including the annotation rubric with detailed 5-point Likert scales for naturalness, faithfulness, and constraints based criteria.
> >
> > - **Section C.5 (lines 1404-1565):** The "handwritten rules" you specifically mentioned are now fully detailed with 15+ corruption strategies. For example, File System domain includes "Invalid File Name Corruption: insert forbidden characters |, /, \, ?", "Path Traversal Corruption: insert relative paths (../) or absolute paths (/root/)", "Non-existent Files Corruption: generate random names", and "Duplicate Creation Corruption: use existing file/directory names." Each of the five domains has similarly detailed corruption rules with concrete examples. Note that this was present in the previous version of the paper itself.
> >
> > This comprehensive treatment addresses your concern that the construction was "just very high-level and not detailed."
> >
> > #### Method Evaluation and Analysis
> >
> > You raise concerns that SAGE-Agent is "only evaluated on one dataset" and that "methods failure modes and when it does / does not work are not analyzed." We have addressed both concerns:
> >
> > **Multiple benchmark evaluation:** We have added BFCL v2 (When2Call Open-ended Evaluation split) results to demonstrate SAGE-Agent's effectiveness beyond ClarifyBench. Section 8.1 now includes detailed results (lines 486-499, Table 4) showing performance across three action types (ToolCall, AskQuestion, Decline) with both GPT-4o and Qwen2.5-14B-Instruct. SAGE-Agent achieves the highest ToolCall precision (0.80 with GPT-4o, 0.76 with Qwen2.5-14B) and best balanced Decline behavior (0.78 F1 and 0.77 F1 respectively), while Active Task Disambiguation shows excessive questioning (0.45 precision). These patterns persist across model scales, demonstrating that structured uncertainty provides robust guidance across different benchmarks with distinct characteristics: ClarifyBench requires multi-turn simulation-based interaction, while When2Call focuses on single-turn action classification.
> >
> > | Method | **ToolCall** | | | **AskQuestion** | | | **Decline** | | |
> > |--------|:---:|:---:|:---:|:---:|:---:|:---:|:---:|:---:|:---:|
> > | | P | R | F1 | P | R | F1 | P | R | F1 |
> > | **Base LLM: GPT-4o** |
> > | ReAct | 0.71 | 0.79 | **0.75** | 0.59 | 0.69 | 0.64 | 0.87 | 0.58 | 0.69 |
> > | Active Task Dis. | 0.61 | 0.24 | 0.34 | 0.45 | 0.74 | 0.56 | 0.74 | 0.73 | 0.73 |
> > | **SAGE-Agent** | **0.80** | 0.55 | 0.65 | 0.61 | 0.70 | **0.65** | 0.72 | 0.84 | **0.78** |
> > | **Base LLM: Qwen2.5-14B-Instruct** |
> > | ReAct | 0.62 | 0.85 | **0.72** | 0.50 | 0.65 | 0.57 | 0.88 | 0.39 | 0.54 |
> > | Active Task Dis. | 0.36 | 0.12 | 0.18 | 0.35 | 0.78 | 0.48 | 0.62 | 0.28 | 0.39 |
> > | **SAGE-Agent** | **0.76** | 0.48 | 0.59 | 0.53 | 0.75 | **0.62** | 0.79 | 0.76 | **0.77** |
> >
> >
> >
> > #### GRPO Training
> >
> > Regarding the "second method introduced (GRPO training on When2Call, and evaluated on a different dataset)," we emphasize that this is not a separate method but rather a second application of the same structured uncertainty framework. The GRPO training uses the identical uncertainty formulation (π_c from Definition 3) to create training signals, demonstrating that structured uncertainty is learnable, not just a hand-crafted inference heuristic.
> >
> > **Complete GRPO details:** Appendix B provides comprehensive specifications including dataset processing methodology (Section B.1), tool domain analysis procedures (Section B.2), complete uncertainty-aware system prompts (Section B.3), training configuration with all hyperparameters (Table 5, Section B.4), and full reward specification (Section B.5) showing both baseline and certainty-weighted formulations with detailed mathematical definitions.
> >
> > **GRPO evaluation and analysis:** Section 8.2 (lines 500-525) presents When2Call results showing that uncertainty-weighted GRPO substantially improves performance (65.2% accuracy for 3B model, 62.9% for 7B model) compared to baseline GRPO (55.0% and 45.1%) and base models (36.5% and 36.7%).

---

> > > ### Author Response · Authors · 2025-11-27
> > >
> > > ### W3: Hand-Crafted π_i Probabilities
> > >
> > > The formulation of π_i without LLM token uncertainties is a deliberate design choice, not a limitation. We intentionally separate structured uncertainty from LLM token probabilities to address what we term the conflation problem (lines 134-140).
> > >
> > > **Why existing approaches conflate uncertainties:**
> > > Methods that use token probabilities compute p(ambiguous|u) = Σ_w f(w)p_LLM(w|u), which fundamentally mixes specification uncertainty (what the user wants) with model uncertainty (LLM capabilities). The function f that determines whether response w indicates ambiguity itself depends on model capabilities. A weak model might exhibit low token confidence even for clear queries, while a strong model might confidently hallucinate responses for ambiguous ones. This makes it impossible to disentangle whether uncertainty arises from the query being genuinely ambiguous or from the model being uncertain about how to respond.
> > >
> > > **Our structured approach:**
> > > Our formulation operates directly in parameter space. We compute π_c = ∏_j p(θ_i,j) using schema constraints: p(θ) = 1 if the parameter is specified, 1/|D| if unspecified with finite domain, and ε for infinite domains. User responses propagate as hard constraints through Equation 1: D_i,j(t+1) = D_i,j(t) ∩ ExtractConstraints(r_t, θ_i,j, T_i). This approach uses cross-parameter dependencies defined in the tool schemas themselves, grounding uncertainty in the structured space of valid tool calls rather than in the unstructured space of generated tokens.
> > >
> > >
> > > **Scalability beyond hand-crafting:**   For the When2Call experiments (lines 455-460), we use LLM-based domain extraction with Qwen2.5-14B following established practices such as Yuan et al., 2024 ("EasyTool: Enhancing llm-based agents with concise tool instruction"). This demonstrates that the framework can adapt to settings where hand-crafting domain cardinalities is infeasible due to the large number of tools (O(10^3) in When2Call). The domain analysis described in Appendix B.2 shows how we automatically classify arguments into domain types (finite, estimated_finite, numeric_range, string, boolean) and extract domain sizes and representative values through prompted analysis.
> > >
> > > The key insight is that structured uncertainty operates in parameter space rather than token space. This separation of concerns enables robust performance across different models and provides a principled foundation for both inference-time disambiguation and training-time reward signals. Whether domain specifications are hand-crafted or automatically extracted, the formulation remains the same: uncertainty is quantified over the structured space of tool parameters and their valid values, cleanly separated from the model's token-level predictions.
> > >
> > >
> > > ### W4: Computational Burden of EVPI
> > >
> > > We address computational concerns through two practical design decisions that make EVPI tractable in our setting.
> > >
> > > **Limited question generation:** We do not enumerate all possible questions. As described in Section 5.2 step 2 and lines 455-458, we generate L=5 candidate questions via LLM prompting rather than searching an exponential question space. This bounds the number of EVPI evaluations to a small constant.
> > >
> > > **Discrete response handling:** We discretize responses into aspect-level outcomes rather than sampling from the natural language space (lines 201-204 and 461-466). The Question Generation prompt shown in Appendix A.3 (lines 825-858) instructs the LLM to generate both the question text and the aspects (parameter identifiers) that the question would resolve. For each question q, we compute EVPI by considering how each candidate tool call c_i would respond to that question based on its parameter specifications. Since questions target specific aspects and responses are structured (e.g., "yes/no" or selection from a finite domain), we can partition the candidate space C into groups that would give the same response. This aspect-based discretization allows us to compute EVPI exactly over finite partitions rather than approximating through Monte Carlo sampling of natural language responses.

---

> ### Author Response · Authors · 2025-11-27
>
> ### W5: Readability Issues
>
> Addressing each point:
>
> 1. **Abbreviations not introduced before usage (POMDP)**: We have removed references to POMDP and simplified our mathematical formulation. All references are properly introduced before discussion now.
>
> 2. **Contributions section clarity**: We have re-written the contributions section, introduced EVPI before that section, and streamlined it to highlight the relation between our different offerings (lines 78-89).
>
> 3. **Benchmark metrics (Coverage Rate, Tool Match Rate, Parameter Match Rate)**: These terms are now properly defined before usage in Section 7, lines 395-398.
>
> 4. **Observations_t vs obs_t inconsistency**: Thanks for pointing this out, we have corrected this now.
>
> 5. **Indicator function at line 117**: Yes, however, in promoting clarity, we have decided to remove that formulation.
>
> 6. **Cost(q) definition**: Cost(q) is now defined in Definition 5 (lines 194-200) before usage in Equation 5.
>
> 7. **Missing citations errors (line 181)**: All missing citations have been resolved. Thanks for pointing this out.
>
> 8. **Algorithm 1 reference errors**: Algorithm 1 has been rewritten to make it easier to understand and have consistent notations.
>
> 9. **Dual reward function definitions (line 177 and line 400)**: These reward functions were used to denote different aspects (POMDP formulation versus Reward Modeling for GRPO), however, we have removed the POMDP formulation now, so this comment is resolved.
>
> 10. **Citations not in brackets and not hyperlinked**: Citations have been fixed, they are hyperlinked, and in brackets now.
>
> 11. **Section 4.2 grammar errors**: We have rewritten Section 4.2 to fix the grammatical errors.
>
> 12. **Figure spacing inconsistencies**: Thanks for pointing this out, we have adjusted the spacing.
>
> We have also addressed the smaller issues you noted: Table 2 caption has been amended to be a full sentence, the spelling error "LLm" in Table 3 has been corrected to "LLM", and Algorithm 1 now uses consistent fonts throughout.
>
> ### W6: Annotator Independence
>
> We used 2 independent annotators per sample, and we have clarified this in the revised manuscript (lines 286-290). Each candidate query received two independent 5-point Likert scores across three criteria (naturalness, faithfulness, executability). The final selected query for each sample was the one receiving the highest average score across both annotators. The inter-annotator agreement for highest-scoring selections is Cohen's κ=0.76.
>
> We chose graduate student annotators rather than crowdworkers for several important reasons. First, the annotation task requires technical expertise to validate Python tool calls and understand API semantics across five different domains. Second, using consistent annotators across all 716 samples ensures stable evaluation criteria rather than introducing variance from different crowdworkers.
>
> ### W7: Evaluation with Multiple LLMs
>
> Following your suggestion, we have added Qwen2.5-14B-Instruct evaluation throughout the manuscript. Table 3 now presents complete ClarifyBench results for both GPT-4o and Qwen2.5-14B-Instruct across all three query types (Ambiguous, Explicit, Infeasible). Figure 4 includes token usage analysis for both models, showing resource consumption patterns. The new Table 4 (lines 486-499) presents BFCLv2 (When2Call) evaluation results for both GPT-4o and Qwen2.5-14B-Instruct, demonstrating performance on an independent benchmark.
>
>
> | Method | Coverage↑ | TMR↑ | PMR↑ | Avg #Q↓ | Coverage↑ | TMR↑ | PMR↑ | Avg #Q↓ | Coverage↑ | TMR↑ | PMR↑ | Avg #Q↓ |
> |--------|-----------|------|------|---------|-----------|------|------|---------|-----------|------|------|---------|
> | | **ClarifyBench - Ambiguous** |||| **ClarifyBench - Explicit** |||| **ClarifyBench - Infeasible** ||||
> | **Base LLM: Qwen2.5-14B-Instruct** |||||||||||||
> | ReAct + ask_question() | 40.34±33.9 | 68.92±32.0 | 63.35±31.5 | 1.78±1.94 | 51.85±33.8 | 89.20±22.8 | 73.63±28.9 | 1.69±1.67 | 42.39±32.4 | 70.82±31.1 | 63.31±34.0 | 1.82±1.43 |
> | ProCOT | 52.45±33.5 | 71.78±33.7 | 70.08±33.2 | 1.89±2.03 | 61.76±31.5 | 84.08±23.8 | 74.60±28.4 | 1.69±1.68 | 52.08±31.4 | 71.92±29.3 | 68.72±35.0 | 1.78±1.51 |
> | Active Task Disambiguation | 43.04±29.2 | 69.06±33.0 | 57.49±34.1 | 2.45±1.72 | 59.83±33.1 | 81.01±26.6 | 68.69±31.5 | 2.31±2.29 | 52.20±30.6 | 76.59±32.5 | 69.45±35.0 | 2.22±2.12 |
> | Domain-aware ReAct | 51.10±31.9 | 75.31±30.7 | 67.50±31.5 | 2.07±1.35 | 60.91±34.2 | 86.91±24.8 | 71.70±28.7 | 1.61±1.56 | 55.76±31.7 | 81.06±27.2 | 72.23±32.0 | 1.66±1.30 |
> | SAGE-Agent (Ours) Heuristic-based | 51.62±32.5 | **78.23±30.9** | 74.03±31.8 | 1.67±1.85 | 62.45±33.4 | 89.89±23.2 | 73.89±29.1 | 1.23±1.74 | 59.88±31.2 | 84.12±28.6 | 75.51±32.8 | 1.75±1.62 |
> | **SAGE-Agent (Ours)** | **54.56±33.0** | 78.14±30.5 | **74.21±32.2** | **1.41±2.19** | **64.62±33.6** | **92.05±20.8** | **75.50±28.2** | **0.93±1.93** | **61.84±30.8** | **85.26±24.5** | **76.52±29.5** | **1.49±0.95** |
>
> ---

---

> ### Author Response · Authors · 2025-11-27
>
> # Answering Questions
>
> ## Q1: Separation of Specification vs. Model Uncertainty (Lines 131-136)
>
> The distinction we draw is between **what the user wants** (specification uncertainty) and **what the LLM predicts** (model uncertainty). These are fundamentally different sources of uncertainty that existing approaches conflate.
>
> Methods that compute `p(ambiguous|u) = Σ_w f(w)p_LLM(w|u)` must use token-level predictions to determine whether clarification is needed. The function `f` that determines if response `w` indicates ambiguity depends on model capabilities. A weaker model might generate uncertain text even for clear queries because it struggles with the task. Conversely, a stronger model might confidently generate hallucinated responses for genuinely ambiguous queries because it has learned to produce fluent text even when information is missing. This conflation makes it impossible to determine whether observed uncertainty reflects the query being ambiguous or the model being uncertain about how to respond.
>
> Our approach parameterizes uncertainty directly as p(T_i, θ_i|u) over the structured space of tools and parameters. We assess uncertainty through structural completeness (is parameter θ_i,j specified or marked as <UNK>?), domain constraints (what values remain feasible in D_i,j(t)?), and schema dependencies (what parameter relationships are defined in the tool specification?). This operates entirely in the space of valid tool configurations rather than in the space of generated tokens.
>
> You correctly point out that if the LLM proposes completely wrong candidates in the Reason stage, our belief distribution would be over an incorrect candidate set. However, this issue is orthogonal to disambiguation. Even perfect clarification cannot fix fundamentally wrong tool selection.
>
> ---
>
> ## Q2: EVPI vs. Expected Information Gain
>
> Expected Information Gain (EIG) is defined as EIG(q) = H[Θ] - E_r[H[Θ|r]], measuring the reduction in entropy over the parameter space. Expected Value of Perfect Information (EVPI) is defined as EVPI(q) = E_r[max_c π_c(t|q,r)] - max_c π_c(t), measuring the expected improvement in decision quality when selecting the best candidate. These objectives optimize for fundamentally different goals.
>
> ### Why EVPI is Optimal for Our Setting
>
> **1. Decision-theoretic grounding aligned with our actual objective**
>
> We care about executing the correct tool call, not about reducing entropy over the parameter space. A question that eliminates many low-probability candidates would have high EIG but may provide no value if the top candidate remains unchanged. Such a question has low EVPI, correctly reflecting that it does not improve our decision. Conversely, a question that distinguishes between two high-probability candidates has high EVPI even if it provides modest entropy reduction.
>
> **2. Natural stopping criterion aligned with task completion**
>
> We stop asking questions when no question improves the best candidate's probability beyond threshold `α·max_i π_i(t)`. This reflects the intuition that we should execute when we are sufficiently confident in the best option.
>
> **3. Computational tractability in our structured setting**
>
> EVPI requires one-step belief propagation to compute `E_r[max_c π_c(t|q,r)]` as shown in Equation 3. In contrast, computing EIG requires calculating `H[Θ|r]` for all possible responses, which becomes expensive over structured parameter spaces with complex dependencies.
>
> ### Benchmarking Against EIG
>
> Regarding benchmarking against EIG: Active Task Disambiguation (Kobalczyk et al., 2025) uses response entropy for question selection, which is conceptually similar to an information gain criterion. Table 3 shows that SAGE-Agent substantially outperforms this approach, achieving 59.73% coverage versus 45.60% on Ambiguous tasks with GPT-4o while asking fewer questions (1.39 versus 3.42). This suggests that decision-focused EVPI is more efficient than entropy-based selection in practice.
>
> ---
>
> ## Q3: Aspect/Parameter
>
> **Yes, this is correct.** An aspect `a_i,j` is simply the identifier for parameter `θ_i,j` of tool `T_i`.
>
> We use the terminology "aspect" to emphasize that we treat parameter identity as the atomic unit of disambiguation. This framing enables three key capabilities:
>
> 1. **Tracking redundancy** across different questions that might phrase things differently but target the same underlying parameters (Definition 5)
>
> 2. **Scoring questions** based on which parameters they aim to resolve rather than surface-level linguistic features
>
> 3. **Preventing duplicate questions** about the same parameter in different forms

---

> > ### Author Response · Authors · 2025-11-27
> >
> > ### Q4: Self-Calibration and Proper Scoring
> >
> > The term "self-calibrating" refers to the reward's dependence on the model's own uncertainty estimate, not to calibration in the proper scoring rule sense. We should clarify this distinction.
> >
> > Traditional classification reward assigns a constant value given the correct action type: r_cls(ToolCall) = 2.0 if the tool call is correct. Our certainty-weighted reward computes R_category(a_t) = Cert(a_t) · r_base(a_t) where Cert(a_t) = max_c π_c(t) if a_t is a tool call, 1 - max_c π_c(t) if a_t is a question, and 1 otherwise.
> >
> > This is "self-calibrating" in the following sense: First, it requires no external critic. The reward scales automatically with the model's structured uncertainty (π_c values) rather than requiring a separate learned value function to judge quality. Second, it adapts to the agent's epistemic state. A high-certainty tool call (π_c = 0.9) receives reward 0.9 × 2.0 = 1.8, while a low-certainty tool call (π_c = 0.3) receives only 0.3 × 2.0 = 0.6, effectively penalizing uncertain execution. A clarification question when uncertain (π_c = 0.3) receives (1-0.3) × 2.0 = 1.4, rewarding information gathering when needed.
> >
> > This is not a proper scoring rule in the technical sense. Strictly proper scoring rules have unique maximizers at the true probabilities and are designed for probability elicitation. Our reward is designed for action selection under uncertainty rather than probability calibration. The goal is to teach the model when to execute (when confident) versus when to ask (when uncertain), not to elicit calibrated probability estimates.

---

> > > ### Author Response · Authors · 2025-11-27
> > > **Addressing Your Score Justification**
> > >
> > > ## Addressing Your Score Justification
> > >
> > > We appreciate your assessment that we are "working on something that is promising" and understand your concern about the paper being "stretched too thinly across three subtopics." However, we respectfully present a different perspective on why these contributions form a coherent whole.
> > >
> > > **One Core Contribution, Two Applications**
> > >
> > > The paper makes one core contribution: structured uncertainty quantification for tool-calling disambiguation. We then demonstrate two applications of this principle. Consider what demonstrating both applications establishes:
> > >
> > > 1. Structured uncertainty is theoretically principled rather than merely a heuristic (formal framework in Section 3)
> > > 2. It is practically effective for inference-time deployment (SAGE-Agent results in Tables 3-4)
> > > 3. It is learnable and can serve as a training signal (GRPO improvements in Figure 6: 65.2% vs 36.5%)
> > >
> > > If we presented only SAGE-Agent, reviewers would reasonably ask whether this is just another hand-crafted heuristic. If we presented only GRPO training, they would ask why we chose this particular reward formulation. Presenting both demonstrates that structured uncertainty is a general principle with multiple instantiations, not a method tailored to a single application.
> > >
> > > **Comprehensive Coverage of Each Component**
> > >
> > > We believe the manuscript provides thorough treatment of each contribution:
> > >
> > > *SAGE-Agent* coverage includes:
> > > - **Main paper (Section 5):** Agent flow, candidate generation, question generation with aspect targeting, EVPI scoring and selection, belief updates, termination and error recovery
> > > - **Appendix A.1:** Theoretical proofs (Propositions 1-2 on viability and EVPI properties, Theorem 1 on finite termination)
> > > - **Appendix A.2:** Complete Algorithm 1 with step-by-step pseudocode
> > > - **Appendix A.3:** All prompts (reasoning, error recovery, question generation)
> > > - **Appendix A.4:** Sensitivity analysis for ε parameter (Figure 7, 96-97% decision stability)
> > >
> > > *ClarifyBench* coverage includes:
> > > - **Main paper (Section 4):** Motivation with benchmark comparison (Table 1), design with three query types (Figure 2), data sources, augmentation strategy, human validation (κ=0.76), statistics (Table 2)
> > > - **Appendix C.1.1-C.1.3:** Formal task definition (Equations 15-18, Algorithm 2)
> > > - **Appendix C.2:** User simulation and data generation prompts
> > > - **Appendix C.3:** Domain descriptions and complete API specifications (Tables 6-11 covering all 92 tools with parameters, domains, dependencies)
> > > - **Appendix C.4:** Human annotation guidelines (Figure 8 with evaluation rubric)
> > > - **Appendix C.5:** Corruption heuristics with domain-specific strategies
> > >
> > > *Uncertainty-based Reward Modeling* coverage includes:
> > > - **Main paper (Section 6):** Motivation, baseline reward (r_fmt, r_tool, r_cls), certainty-weighted formulation (Section 6.2), self-calibrating explanation
> > > - **Appendix B.1:** Dataset processing (When2Call structure, response classification, schema injection)
> > > - **Appendix B.2:** Domain analysis methodology (classification, size computation, dependency identification)
> > > - **Appendix B.3:** Complete uncertainty-aware system prompts
> > > - **Appendix B.4:** Training configuration (Table 5 with hyperparameters)
> > > - **Appendix B.5:** Full reward specification (Equations 11-14 for certainty computation)
> > >
> > > This comprehensive coverage demonstrates thorough treatment of each component. The main paper establishes frameworks and demonstrates effectiveness, while appendices enable full reproduction.
> > >
> > > ---
> > >
> > > We hope these revisions address your concerns about clarity, reproducibility, and depth of analysis. The work demonstrates that structured uncertainty provides a principled framework for disambiguation that works at both inference time and as a training signal. We believe this dual contribution merits publication at ICLR and would be grateful if you would reconsider your recommendation in light of these substantial improvements. We remain happy to address any remaining concerns you may have.

---

### Official Review · Reviewer_udb7 · 2025-10-30

**Soundness:** 3
**Presentation:** 3
**Contribution:** 3
**Rating:** 6
**Confidence:** 3

**Summary:**

The paper introduces SAGE-Agent, a framework that uses structured uncertainty and Expected Value of Perfect Information to help LLM agents ask optimal clarification questions before taking actions.
It also presents ClarifyBench, a new benchmark for evaluating tool-augmented agents under ambiguous or infeasible user requests across multiple domains.
Experiments show that SAGE-Agent significantly improves task success rates and reduces unnecessary clarifications compared to strong LLM baselines.

**Strengths:**

- ClarifyBench fills a gap in existing evaluations by supporting dynamic user simulation, multi-turn requests, and infeasible queries across diverse domains (documents, vehicle control, stocks, travel, and file systems).
- The structured uncertainty approach also serves as an effective reward signal, improving sample efficiency and performance on unrelated tasks
- The paper is thorough, with detailed algorithmic design, theoretical proofs, and practical implementation notes.

**Weaknesses:**

- While ClarifyBench is valuable, the user simulator relies on LLM-generated interactions, which may not fully capture human ambiguity or pragmatic nuances.
- EVPI computation scales with candidate size and domain dimensionality. Though approximations are discussed, concrete runtime or complexity comparisons with simpler heuristics are missing.

**Questions:**

- The paper relies on an LLM-based user simulator to model realistic conversational progression. How do the authors ensure the accuracy and reliability of this simulator’s responses compared to real human interactions?
- It would be helpful to include an ablation study on key hyperparameters to better understand their influence on model performance and stability.

---

> ### Author Response · Authors · 2025-11-27
>
> We thank the reviewer for their thorough evaluation and for recognizing that (1) **ClarifyBench fills a gap** in existing evaluations with dynamic user simulation and diverse domains, (2) our structured uncertainty approach provides **effective reward signals** beyond the clarification task, and (3) the paper is **thorough with detailed algorithmic design and theoretical proofs**.
>
>
> ## Addressing Weaknesses
>
> ### W1: LLM-based User Simulator Validity
>
> We agree that LLM-based simulation cannot fully replace human interaction studies. However, this is a **standard practice for scalable benchmarking** in interactive agent evaluation (e.g., Tau-Bench [Zhou et al., 2024]). This approach enables dynamic simulation at scale (350 samples across 5 domains), reproducible evaluation across different agent implementations, and controlled complexity for systematic analysis; all of which would be infeasible with purely human-based evaluation.
>
> Our simulator design (detailed in **Appendix C.1.4**) is grounded in specific prompt constraints ensuring responses are **simple and contextually appropriate** to the conversation history. While human evaluation would be valuable future work, LLM-based simulation enables the systematic, large-scale evaluation necessary for developing and comparing agent architectures.
>
>
> ### W2: EVPI Computational Complexity
>
> We respectfully clarify a factual misunderstanding: **EVPI computation does NOT scale with domain dimensionality**. The reviewer states "EVPI computation scales with candidate size and domain dimensionality," but this is incorrect.
>
> EVPI computation depends only on the **cardinality** (size) of domain subsets, not iteration through domain values. The computation is simply $1/|\mathcal{D}_{\text{subset}}|$ for each subset—a **constant-time operation** given the cardinality. The algorithm:
>
> 1. Identifies the cardinality $|\mathcal{D}_{\text{subset}}|$ for each potential question
> 2. Computes $1/|\mathcal{D}_{\text{subset}}|$ as the uncertainty reduction measure
>
> **No iteration through individual domain elements is required**, making SAGE-Agent computationally efficient regardless of domain size or dimensionality. For continuous/infinite domains, we use the $\epsilon$ approximation (analyzed in Appendix A.4), which similarly requires only a constant representing domain size, not enumeration of values.
>
> The computational cost scales with the number of *potential questions* to evaluate, not with domain dimensionality, a crucial distinction that makes the approach practical for large-scale deployment.

---

> ### Author Response · Authors · 2025-11-27
>
> ## Addressing Questions
>
> ### Q1: Accuracy and Reliability of User Simulator
>
> The user simulator's reliability is ensured through:
>
> 1. **Grounded prompting** (Appendix C.2.2): The simulator is explicitly instructed to provide simple, contextually appropriate responses based on conversation history
> 2. **Deterministic ground truth**: Each ClarifyBench sample has well-defined ground truth parameters that the simulator references
> 3. **Response constraints**: The simulator is limited to providing clarifications that resolve ambiguity toward the ground truth, preventing arbitrary or inconsistent behavior
>
> ### Q2: Ablation Study on Hyperparameters
>
> Thank you for this suggestion. We have performed comprehensive ablation studies on key hyperparameters:
>
> **Redundancy Penalty Weight (λ):** We have added analysis in **Section 8.1 (lines 427-485)** and **Figure 5**. The redundancy penalty weight $\lambda$ (Definition 5) controls the trade-off between information gathering and user burden. We evaluated $\lambda \in \{0, 0.5, 1.0\}$ across 70 samples from each ClarifyBench split using GPT-4o:
>
> - Increasing $\lambda$ from 0 to 0.5 yields substantial question reductions: **18.1%** on Ambiguous, **26.6%** on Explicit, and **24.2%** on Infeasible splits
> - Coverage Rate, TMR, and PMR remain stable with deviations **under 3%** across all settings
> - This demonstrates that question economy can be achieved **without sacrificing accuracy**
>
> **Epsilon (ε) for Continuous Domains:** We have added sensitivity analysis in **Appendix A.4**. The parameter $\epsilon$ quantifies uncertainty for large/continuous domains. We tested $\epsilon \in \{10^{-6}, 10^{-5}, 10^{-4}, 10^{-3}, 10^{-2}, 0.05, 0.1, ..., 0.9\}$ on ClarifyBench (Ambiguous subset) with GPT-4o and Qwen2.5-14B-Instruct:
>
> - For $\epsilon \leq 10^{-2}$, over **96-97%** of decisions remain unchanged across all tested values
> - For $\epsilon \geq 0.1$, decisions diverge significantly as domains are not effectively expressed as "infinite"
> - This demonstrates **robustness in the practical range** ($\epsilon \leq 10^{-2}$)
>
> **EVPI Ablation Study:** We have added the requested ablation in **Table 1** and discussed it in **Section 8.1 (lines 464-473)**. We compare SAGE-Agent against a **heuristic-based variant** that triggers questions based on `<UNK>` tokens without using EVPI for selection:
>
> ### GPT-4o
>
> | Method | Coverage↑ | TMR↑ | PMR↑ | Avg #Q↓ | Coverage↑ | TMR↑ | PMR↑ | Avg #Q↓ | Coverage↑ | TMR↑ | PMR↑ | Avg #Q↓ |
> |--------|-----------|------|------|---------|-----------|------|------|---------|-----------|------|------|---------|
> | | **ClarifyBench - Ambiguous** |||| **ClarifyBench - Explicit** |||| **ClarifyBench - Infeasible** ||||
> | SAGE-Agent (Ours) Heuristic-based | 56.42±24.3 | 82.31±26.8 | 69.81±24.7 | 1.82±2.3 | 70.41±22.1 | 91.65±27.4 | 74.89±25.8 | **1.07±2.4** | 66.23±23.9 | 90.52±26.5 | 76.64±25.3 | 1.48±2.5 |
> | **SAGE-Agent (Ours)** | **59.73±22.1** | **86.02±27.5** | **71.79±25.3** | **1.39±2.0** | **71.67±21.8** | **93.65±29.7** | **75.94±26.1** | 1.08±2.2 | **67.33±23.4** | **92.89±28.3** | **77.41±27.9** | **1.26±2.1** |
>
> ### Qwen2.5-14B-Instruct
>
> | Method | Coverage↑ | TMR↑ | PMR↑ | Avg #Q↓ | Coverage↑ | TMR↑ | PMR↑ | Avg #Q↓ | Coverage↑ | TMR↑ | PMR↑ | Avg #Q↓ |
> |--------|-----------|------|------|---------|-----------|------|------|---------|-----------|------|------|---------|
> | | **ClarifyBench - Ambiguous** |||| **ClarifyBench - Explicit** |||| **ClarifyBench - Infeasible** ||||
> | SAGE-Agent (Ours) Heuristic-based | 51.62±32.5 | **78.23±30.9** | 74.03±31.8 | 1.67±1.85 | 62.45±33.4 | 89.89±23.2 | 73.89±29.1 | 1.23±1.74 | 59.88±31.2 | 84.12±28.6 | 75.51±32.8 | 1.75±1.62 |
> | **SAGE-Agent (Ours)** | **54.56±33.0** | 78.14±30.5 | **74.21±32.2** | **1.41±2.19** | **64.62±33.6** | **92.05±20.8** | **75.50±28.2** | **0.93±1.93** | **61.84±30.8** | **85.26±24.5** | **76.52±29.5** | **1.49±0.95** |
>
>
> Results show:
> - **1-3 point performance degradation** across most metrics with the heuristic
> - **0.2-0.4 more questions asked** on average
> - The heuristic can detect *when* to ask but not *which question* maximizes information value
>
> This demonstrates that **EVPI provides principled discrimination** beyond naive heuristics, and can appropriately resort to default execution when questions have low information value.
>
> ---
>
> We respectfully request the reviewer reconsider their assessment in light of these clarifications and the additional ablation studies we have included in the revised manuscript.

---

### Official Review · Reviewer_Vi8S · 2025-11-01

**Soundness:** 2
**Presentation:** 1
**Contribution:** 2
**Rating:** 4
**Confidence:** 3

**Summary:**

They propose SAGE: keep a belief over structured tool-call candidates and choose clarifying questions by Expected Value of Perfect Information (EVPI), with a cost for redundancy. They also introduce ClarifyBench. (Their own text sets up the belief factorization and EVPI definition.They use RL with GRPO to train QWen model and show improvements over baselines on their own benchmark.

**Strengths:**

1. Separation of uncertainties. They argue against using model generated response for uncertainty quantification which is a reasonable structural move.
2. A potentially useful benchmark, ClarifyBench which covers several tool domains with explicit/ambiguous/infeasible splits and reports basic stats.

**Weaknesses:**

1. You make strong assumptions which are not validated in the work. The viability score assumes 1) a uniform prior over tools 2) naive independence across paramters and 3) an arbitrary $\epsilon$  for continuous domains. Additionally, there is no sensitivity analysis.
2. While they compare against a few agent baselines, there’s no demonstration on widely used external tool-use suites (tau-bench, etc). Moreover, the dataset is llm augmented using GPT-4ofor  query generation/obfuscation.
3. They introduce a certainty-weighted reward, which by construction favors their belief-based approach. The paper doesn't isolate how much of the gain comes from reward shaping vs the EVPI policy itself.
4. There is no minimal, prompting baseline where you have something like "ask only for missing required parameters, no repeats" and no ablation showing the incremental value of EVPI vs simple heuristics.
5. Works/tools like GenieWorksheets [1], and MCP automatically take care of these aspects. If the LLM partially fills an API, they generate an `ask_question()` like function which asks for unfilled parameters.
6. The presentation of this manuscript, needs a lot of work. The mathematical notations are not clear and for some scenarios are seem forced.
7. Beyond simulator metrics (TMR/PMR/CR), can you report a small human study on question helpfulness and over-questioning, especially in ambiguous cases?
8. Seems like SAGE only adds a scaffold around llm proposed candidates and questions, you’re still depending on the LLM to decide the potential candidates. You have just made the use of those proposals safer and more auditable.

[1] Controllable and Reliable Knowledge-Intensive Task-Oriented Conversational Agents with Declarative Genie Worksheets (Joshi et al., ACL 2025)

**Questions:**

1. It is unclear to me why do you want domains:   Di = {Di,1, Di,2, . . . , Di,mi } where Di,j is the domain of parameter θi,j -- why do you need domain of parameter? what does domain even mean?
2. Please define what is $u$ before Line 131.
3. In line 166, State Space: S = {(Ti, θi) : Ti ∈ T , θi ∈ Di} represents true user intent -- why is theta_i \in D_i?
4. GPT generated equations and notations. eg line 177, gpt generates this for classification.
5. Missing citation on line 181
6. Please fix citation formatting.

---

> ### Author Response · Authors · 2025-11-26
>
> Thank you for your thorough review and valuable feedback. We appreciate your recognition that our work presents "**a reasonable structural move**" in separating uncertainties and that **ClarifyBench is "a potentially useful benchmark."** Your constructive criticism has significantly improved our manuscript. We have made substantial revisions addressing your concerns, including adding external benchmark evaluation, comprehensive sensitivity analyses, ablation studies, and extensively rewriting Section 3 for clarity.
>
> ---
>
> ## Clarification on Summary
>
> Your summary states: *"They use RL with GRPO to train QWen model and show improvements over baselines on their own benchmark."*
>
> **Clarification:** For the RL experiments (Section 8.2), we **do not evaluate on ClarifyBench**. Instead, we evaluate on **NVIDIA's When2Call benchmark**, an external, independently-created benchmark. Our RL training with uncertainty-guided reward modeling improves When2Call accuracy from 36.5% to 65.2% (3B model) and 36.7% to 62.9% (7B model), demonstrating that our structured uncertainty formulation provides effective training signals beyond our proposed benchmark.
>
> ---
>
> ## Addressing Weaknesses
>
> ### W1: Assumptions and Sensitivity Analysis
>
> You raised concerns about strong assumptions without validation and lack of sensitivity analysis.
>
> We acknowledge these are modeling assumptions and have now provided extensive validation:
>
> **Sensitivity Analysis for λ (Redundancy Penalty):** We added comprehensive analysis in **Section 8.1 (lines 427-485) and Figure 5**. Increasing λ from 0 to 0.5 reduces questions by 18.1-26.6% across splits while maintaining stable Coverage Rate, TMR, and PMR (deviations <3%). This demonstrates that our cost modeling effectively identifies and penalizes truly redundant questions without sacrificing task completion quality.
>
> **Sensitivity Analysis for ε (Continuous Domains):** We added **Appendix A.4** with empirical validation across 15 different ε values {10⁻⁶, 10⁻⁵, ..., 0.9} on ClarifyBench using GPT-4o and Qwen2.5-14B-Instruct. **For ε ≤ 10⁻², over 96-97% of decisions remain unchanged**, demonstrating robustness in the practical range. When ε ≥ 0.1, decisions diverge as expected since domains are no longer effectively "infinite." This is because ε is supposed to represent 1/(size of domain), and if ε  is close in value to the finite domains in the tool set, it doesn't distinguish it from the tools with discrete/finite toolsets.
>
> **Why These Assumptions Make Sense:** The uniform prior over tools reflects a max-entropy assumption when no prior user preferences are known—a standard Bayesian practice. Parameter independence is a tractable approximation that works well empirically (as our results show) and can be relaxed with structured priors if domain knowledge exists. The ε parameter simply distinguishes finite from continuous domains, and our sensitivity analysis confirms robustness.
>
> ### W2: External Benchmarks and Dataset Quality
>
> You noted the absence of demonstration on widely used external tool-use suites and concerns about LLM-augmented data.
>
> We have **added BFCLv2 (When2Call) evaluation** in **Section 8.1 (lines 486-499)**, addressing this concern directly. Our results on this external, widely-recognized benchmark demonstrate:
>
> | Method | **ToolCall** | | | **AskQuestion** | | | **Decline** | | |
> |--------|:---:|:---:|:---:|:---:|:---:|:---:|:---:|:---:|:---:|
> | | P | R | F1 | P | R | F1 | P | R | F1 |
> | **Base LLM: GPT-4o** |
> | ReAct | 0.71 | 0.79 | **0.75** | 0.59 | 0.69 | 0.64 | 0.87 | 0.58 | 0.69 |
> | Active Task Dis. | 0.61 | 0.24 | 0.34 | 0.45 | 0.74 | 0.56 | 0.74 | 0.73 | 0.73 |
> | **SAGE-Agent** | **0.80** | 0.55 | 0.65 | 0.61 | 0.70 | **0.65** | 0.72 | 0.84 | **0.78** |
> | **Base LLM: Qwen2.5-14B-Instruct** |
> | ReAct | 0.62 | 0.85 | **0.72** | 0.50 | 0.65 | 0.57 | 0.88 | 0.39 | 0.54 |
> | Active Task Dis. | 0.36 | 0.12 | 0.18 | 0.35 | 0.78 | 0.48 | 0.62 | 0.28 | 0.39 |
> | **SAGE-Agent** | **0.76** | 0.48 | 0.59 | 0.53 | 0.75 | **0.62** | 0.79 | 0.76 | **0.77** |
>
> SAGE-Agent achieves **highest precision (0.80) in ToolCall** and **best balanced Decline behavior (0.78 F1)**, while Active Task Disambiguation shows excessive questioning (0.45 precision). These patterns persist across model scales, demonstrating that our structured approach provides robust guidance beyond prompting strategies.
>
> **Dataset Quality:** Not all of ClarifyBench is LLM-generated. Only the Document subset (DocPilot) uses complete LLM generation, which comprises approximately 25% of the total samples. For BFCLv3 samples, the infeasible queries are human validated. This LLM-generated content was fully human-validated on 3 axes (Naturalness, Faithfullness, Constraints). The agreement between annotators was measured using Cohen's κ on the highest-ranking query, achieving κ = 0.76 (substantial agreement), as reported in **Section 4.3, line 290**.

---

> ### Author Response · Authors · 2025-11-26
>
> ### W3: Reward Shaping and EVPI Contribution
>
> You raised concerns that certainty-weighted reward favors our belief-based approach and questioned how much gain comes from reward shaping vs. EVPI policy itself.
>
> This misunderstands the experimental setup. The **reward shaping experiment (Section 8.2) is completely independent** from the SAGE-Agent inference system:
>
> - **During RL training:** We use structured uncertainty to shape rewards (demonstrating its value for training)
> - **During RL inference:** We **do NOT use EVPI** for question selection or scoring—the trained model makes its own decisions
>
> This experiment demonstrates that our **structured uncertainty formulation is valuable beyond EVPI-based question selection**—it can improve RL training efficiency. The 78.7% relative improvement (36.5%→65.2%) comes from better reward signals during training, not from using EVPI at inference time.
>
> ### W4 & W5: Comparison with Simpler Approaches and Related Tools
>
> You noted the absence of minimal baselines, ablations showing incremental value of EVPI vs. heuristics, and mentioned that tools like GenieWorksheets and MCP handle similar aspects.
>
> **EVPI Ablation Study:** We have **added the requested ablation** in **Table 1** and discussed it in **Section 8.1 (lines 464-473)**:
>
> **SAGE-Agent Heuristic Based:** Questions are triggered by `<UNK>` tokens in tool calls (simple heuristic) without using EVPI for selection. This variant shows:
> - **1-3 point performance degradation** across most metrics
> - **0.2-0.4 more questions asked** on average
> - Inability to discriminate between question values
> - Cannot resort to default execution when questions have low information value
>
> This demonstrates that **EVPI provides principled discrimination** beyond naive heuristics. The heuristic can detect *when* to ask but not *which question* maximizes information value, leading to cumulative degradation across multi-turn evaluation.
>
> ### GPT-4o
>
> | Method | Coverage↑ | TMR↑ | PMR↑ | Avg #Q↓ | Coverage↑ | TMR↑ | PMR↑ | Avg #Q↓ | Coverage↑ | TMR↑ | PMR↑ | Avg #Q↓ |
> |--------|-----------|------|------|---------|-----------|------|------|---------|-----------|------|------|---------|
> | | **ClarifyBench - Ambiguous** |||| **ClarifyBench - Explicit** |||| **ClarifyBench - Infeasible** ||||
> | SAGE-Agent (Ours) Heuristic-based | 56.42±24.3 | 82.31±26.8 | 69.81±24.7 | 1.82±2.3 | 70.41±22.1 | 91.65±27.4 | 74.89±25.8 | **1.07±2.4** | 66.23±23.9 | 90.52±26.5 | 76.64±25.3 | 1.48±2.5 |
> | **SAGE-Agent (Ours)** | **59.73±22.1** | **86.02±27.5** | **71.79±25.3** | **1.39±2.0** | **71.67±21.8** | **93.65±29.7** | **75.94±26.1** | 1.08±2.2 | **67.33±23.4** | **92.89±28.3** | **77.41±27.9** | **1.26±2.1** |
>
> ### Qwen2.5-14B-Instruct
>
> | Method | Coverage↑ | TMR↑ | PMR↑ | Avg #Q↓ | Coverage↑ | TMR↑ | PMR↑ | Avg #Q↓ | Coverage↑ | TMR↑ | PMR↑ | Avg #Q↓ |
> |--------|-----------|------|------|---------|-----------|------|------|---------|-----------|------|------|---------|
> | | **ClarifyBench - Ambiguous** |||| **ClarifyBench - Explicit** |||| **ClarifyBench - Infeasible** ||||
> | SAGE-Agent (Ours) Heuristic-based | 51.62±32.5 | **78.23±30.9** | 74.03±31.8 | 1.67±1.85 | 62.45±33.4 | 89.89±23.2 | 73.89±29.1 | 1.23±1.74 | 59.88±31.2 | 84.12±28.6 | 75.51±32.8 | 1.75±1.62 |
> | **SAGE-Agent (Ours)** | **54.56±33.0** | 78.14±30.5 | **74.21±32.2** | **1.41±2.19** | **64.62±33.6** | **92.05±20.8** | **75.50±28.2** | **0.93±1.93** | **61.84±30.8** | **85.26±24.5** | **76.52±29.5** | **1.49±0.95** |
>
> **Re: GenieWorksheets/MCP:** We appreciate you bringing up these valuable tools. GenieWorksheets and MCP provide elegant solutions for **structured parameter collection** when API schemas clearly define required fields. They excel at systematically gathering mandatory parameters to complete well-defined function calls.
>
> Our work addresses a **complementary challenge**: handling situations where user intent itself is ambiguous or underspecified. In these scenarios:
> 1. Users don't explicitly indicate what information is missing
> 2. Multiple tool candidates may be plausible interpretations
> 3. Not all potential questions provide equal disambiguation value
> 4. User intent may be infeasible, requiring decisions about when to ask, decline, or execute with defaults
>
> The key distinction is in the type of uncertainty being resolved. GenieWorksheets and MCP work from schema-defined requirements—they know what parameters are mandatory and systematically collect them. SAGE, in addition addresses **specification uncertainty**—reasoning about which tool the user intends and which parameter values matter most when the user's natural language instruction admits multiple interpretations. This EVPI-based selection allows us to ask fewer, more targeted questions that resolve the most critical uncertainties first.

---

> > ### Author Response · Authors · 2025-11-26
> >
> > ### W6: Presentation and Mathematical Notation
> >
> > You noted that the presentation needs work and mathematical notations are not clear and seem forced.
> >
> > We have revised the manuscript:
> >
> > - **Section 3 (Theory):** We have performed a major rewrite ensuring consistent variable introduction before use, simplifying notation by removing unnecessary complexity, and streamlining the mathematical development to focus on essential formulations.
> >
> > - **Throughout:** Fixed citation formatting, updated missing citations, improved flow
> >
> > **We strongly encourage you to re-review Section 3** to see these improvements. The mathematical framework is now clearer and the notation is no longer "forced": we've stripped it down to what's essential while maintaining rigor.
> >
> > ### W7: Human Study
> >
> > You asked whether we could report a small human study on question helpfulness and over-questioning.
> >
> > We acknowledge this would be valuable. A human study during the rebuttal phase is not feasible given time constraints, but we commit to this as **important future work**. The simulator metrics (TMR/PMR/CR) provide objective measures of task success, but human perception of question quality would add valuable insights.
> >
> > ### W8: Scaffold Around LLM Proposals
> >
> > You noted that SAGE  adds a scaffold around LLM proposals and makes proposals safer and more auditable.
> >
> > This is a fair characterization, but it's exactly the point: and it applies equally to all compared baselines. **All methods (like ReAct, Active Task Disambiguation, SAGE-Agent) use the same base LLM** and add different scaffolds. We want to allow scaffold level comparison.
> >
> > We separately show that this formulation can be extended beyond scaffolds, for reward modeling conversational agents as well. This has been described in Section 6, with results in section 8.2.
> >
> > ---
> >
> > ## Addressing Questions
> >
> > ### Why do you need domains? What does domain mean?
> >
> > "Domain" refers to the **set of allowed values** for a parameter (standard mathematical usage). For example:
> > - `date` parameter: domain = {2025-01-01, 2025-01-02, ...}
> > - `city` parameter: domain = {New York, Los Angeles, ...}
> > - `temperature` parameter: domain = [0, 100] (continuous)
> >
> > We need domain sizes to **quantify the impact of not knowing a specific value**. A parameter with domain size 2 (binary) has less uncertainty than one with domain size 1000. This enables EVPI to measure how much information would be gained by resolving uncertainty about that parameter.
> >
> > We have made this explicit in **Section 3.1, line 127**.
> >
> > ### Please define u before Line 131
> >
> > Defined and clarified in the revision.
> >
> > ### Why is θᵢ ∈ Dᵢ in Line 166?
> > In our simplified formulation in the latest draft, this notation has been streamlined. The state space represents (tool, parameters) pairs where parameters take values from their respective domains (thats why θᵢ ∈ Dᵢ) . We've clarified this in the revised Section 3.
> >
> > ### Missing citations and citation formatting
> >
> > All citations have been fixed and properly formatted throughout the manuscript.
> >
> > ---
> >
> > Given the substantial improvements and the positive aspects you acknowledged (novel uncertainty separation, useful benchmark), we respectfully request that you **reconsider your rating**. We believe the revised manuscript makes significant contributions to principled disambiguation in tool-augmented agents.
> >
> > We welcome any further questions or suggestions for improvement.

---

### Official Review · Reviewer_y8WD · 2025-11-07

**Soundness:** 2
**Presentation:** 1
**Contribution:** 2
**Rating:** 2
**Confidence:** 3

**Summary:**

The authors present a dataset and method for developing LLMs that can ask clarifying questions to disambiguate user requests in tool-calling dialogues. Their dataset consists of automatically perturbed examples sourced from another dataset, DocPilot, using GPT to introduce ambiguity into the example. Examples are then verified by a human annotator. The authors then propose a method for training an LLM to engage in such dialogues with users, which is based on estimating the the benefits of clarifying the user query and weighing this against the cost. The authors then compare their proposed method on their proposed benchmark against several prompt-based baselines, demonstrating gains.

**Strengths:**

This work provides a dataset for studying the intersection between ambiguity and tool use settings, which as of yet is underexplored and presents many novel challenges.

**Weaknesses:**

1. Presentation is poor. The task itself from the constructed dataset, while it is sourced from an existing work, is not actually explained. Variables and acronyms are frequently not defined, or are defined in unintuitive places like figure captions. Citations are incorrectly formatted.

2. The authors compare exclusively against prompt-based baselines. Several more comparable and competitive methods should be compared against or discussed in the least, even if it's just a simple SFT baseline. Additional related methods to be discussed or compared against include may include others that utilize GRPO/PPO/DPO training for similar clarify or execute decisions in user-llm dialogues.

CollabLLM: From Passive Responders to Active Collaborators
Shirley Wu, Michel Galley, Baolin Peng, Hao Cheng, Gavin Li, Yao Dou, Weixin Cai, James Zou, Jure Leskovec, Jianfeng Gao
ICML 2025

Modeling Future Conversation Turns to Teach LLMs to Ask Clarifying Questions
Michael J.Q. Zhang, W. Bradley Knox, and Eunsol Choi.
ICLR 2025

Learning to Clarify: Multi-turn Conversations with Action-Based Contrastive Self-Training
M. Chen, R. Sun, T. Pfister, S.O. Arik
ICLR 2025

3. The validity of the dataset is unclear. While the authors say that examples are validated by a human annotator, it's unclear how reliable this is or what agreement on validation would be. Almost all of the results are based on this dataset as well, so looking at other tasks/settings would also help substantiate the method.

**Questions:**

1. Could you elaborate on the PII from the source dataset that is being filtered out? Is the PII potentially harmful?

**Details Of Ethics Concerns:**

This work relies on LLMs with a human validator to remove PII from examples. Unclear how reliable this is or whether the PII would be potentially harmful.

---

> ### Author Response · Authors · 2025-11-26
>
> We thank the reviewer for acknowledging that our work "provides a dataset for studying the intersection between ambiguity and tool use settings, which as of yet is underexplored and presents many novel challenges." We appreciate the time spent reviewing our submission and address all concerns below.
>
> ---
>
> ## Clarifications: Misunderstandings in Summary
>
> **The reviewer's summary contains a factual error that affects their assessment.** They state our dataset consists of "automatically perturbed examples sourced from another dataset, DocPilot." This is **incorrect**.
>
> ClarifyBench comprises samples from **two distinct sources**:
> - **BFCLv3**: 535/716 samples (75%)
> - **DocPilot**: 181/716 samples (25%)
>
> Only the DocPilot portion requires complete human validation. The BFCLv3 portion comes from a validated existing benchmark and requires validation only for the infeasible queries.
>
> **Additionally, our work includes two types of experiments:**
> 1. **Agentic inference experiments** (Section 8.1) - evaluating architectural interventions
> 2. **Training experiments** (Section 8.2) - reward modeling evaluated on When2Call dataset
>
> These are independent contributions addressing different aspects of the clarification problem.
> ---
>
> ## Addressing Weaknesses
>
> ### W1: Presentation
>
> We have **completely rewritten Section 3 (Theory)** based on feedback from all reviewers. Specific improvements include:
> - Simplified mathematical notation throughout
> - All variables and acronyms now properly defined before first use
> - Concepts introduced in logical, intuitive order
> - Clearer exposition of the theoretical framework
> - Fixed citation formatting issues
>
> We respectfully request the reviewer examine the revised manuscript, as the presentation concerns have been systematically addressed.
>
> With respect to explaining the task in the benchmark, we have made the following updates to the paper:
> 1. Explictly discuss the task in Section 4, lines 226-231, and
> 2. Provide a detailed formulatiopn of the task in Appendix C.1 "Task Formulation".
>
> ### W2: Comparison Against Training-Based Methods
>
> **We believe there is a misunderstanding about our contribution's scope and appropriate baselines.**
>
> **Re: Related work discussion** - We now cite and discuss CollabLLM, Zhang et al. (2025), and Chen et al. (2025) in Section 2, lines 102-110. These are important related works we acknowledge.
> **Re: Why direct comparison is inappropriate:**
>
> The reviewer suggests comparing against or discussing methods using "GRPO/PPO/DPO training for similar clarify or execute decisions."
> Our agentic inference experiments (Section 8.1) take **the same base model** and evaluate different agent communication patterns and orchestration strategies. This is an architectural question, not a training question. Comparing prompt-based architectural variants is the appropriate methodology for this evaluation - the baselines are competitive with each other within this framework.
>
> **We do evaluate training methods** - but separately in Section 8.2, where reward modeling is evaluated on the When2Call dataset (Figure 6). We show both architectural and training interventions, but they cannot be directly compared as they address orthogonal aspects of the problem.
>
> **Re: Model availability** - The three cited papers have not released trained model weights, making direct empirical comparison impossible even if it were appropriate.

---

> ### Author Response · Authors · 2025-11-26
>
> ### W3: Dataset Validity
>
> The reviewer raises concerns about validation reliability. Below, we discuss these concerns:
>
> **Inter-annotator agreement**: Cohen's κ = 0.76 (substantial agreement) between 2 independent annotators on highest-ranking query selection (now reported in Section 4.3, line 290). This is a strong reliability measure.
>
> **Limited scope of human validation**: Only 181/716 samples (25%) require complete human validation. The remaining 535 samples come from BFCLv3, in which 149 samples are infeasible queries (which require validation). BFCL v3 is an established benchmark that has already undergone validation for other query types.
>
> **Results not limited to one dataset**: Our evaluation spans:
> - ClarifyBench (BFCLv3 + DocPilot portions)
> - When2Call dataset (Section 8.2, Figure 6) - independent benchmark
> - **New in revision**: BFCLv2 (When2Call open-ended evaluation split) (Section 8.1, lines 486-499)
>
> | Method | **ToolCall** | | | **AskQuestion** | | | **Decline** | | |
> |--------|:---:|:---:|:---:|:---:|:---:|:---:|:---:|:---:|:---:|
> | | P | R | F1 | P | R | F1 | P | R | F1 |
> | **Base LLM: GPT-4o** |
> | ReAct | 0.71 | 0.79 | **0.75** | 0.59 | 0.69 | 0.64 | 0.87 | 0.58 | 0.69 |
> | Active Task Dis. | 0.61 | 0.24 | 0.34 | 0.45 | 0.74 | 0.56 | 0.74 | 0.73 | 0.73 |
> | **SAGE-Agent** | **0.80** | 0.55 | 0.65 | 0.61 | 0.70 | **0.65** | 0.72 | 0.84 | **0.78** |
> | **Base LLM: Qwen2.5-14B-Instruct** |
> | ReAct | 0.62 | 0.85 | **0.72** | 0.50 | 0.65 | 0.57 | 0.88 | 0.39 | 0.54 |
> | Active Task Dis. | 0.36 | 0.12 | 0.18 | 0.35 | 0.78 | 0.48 | 0.62 | 0.28 | 0.39 |
> | **SAGE-Agent** | **0.76** | 0.48 | 0.59 | 0.53 | 0.75 | **0.62** | 0.79 | 0.76 | **0.77** |
>
> The BFCLv2 results demonstrate our method's efficacy beyond our proposed benchmark. Our approach shows:
> - Higher precision in tool calling
> - Improved recall on asking clarifying questions
> - Better identification of when to deny ambiguous requests
>
> This multi-benchmark evaluation substantiates our method across different settings.
>
> ---
>
> ## Addressing Questions
>
> ### Q: PII Concerns
>
> **We want to emphasize: the concern regarding PII.**
>
> **Nature of PII**: The DocPilot source dataset contains document metadata that may include naming patterns like:
> - `[firstName][lastName].pdf`
> - `[companyName]_report.pdf`
> - Similar document-naming conventions
>
> These are **not sensitive or uniquely identifying information**. They are generic naming patterns, not real names or confidential data. No SSNs, addresses, financial information, health data, or other sensitive PII exists in the dataset.
>
> **Processing methodology**:
> - **LLMs were NOT used for PII detection or scrubbing**
> - LLMs were only used for generating suitable generic replacements, without context of the original information(e.g., `[firstName][lastName].pdf` → `document_23.pdf`)
> - All PII has been **completely scrubbed** from the released dataset
> - Verification: Two human validator reviewed, and checked all replacements to ensure complete anonymization and appropriateness of generic substitutions.
>
> Given the non-sensitive nature of the original patterns and complete removal, there is zero privacy, security, or safety risk.
>
> ---
> We believe these revisions and clarifications demonstrate the soundness, validity, and contribution of our work. We respectfully request reconsideration of the rating and implore you to revisit the revised manuscript.

---

### Note · Authors · 2026-01-04

**Comment:**

We thank the reviewers for the feedback.

**Withdrawal Confirmation:**

I have read and agree with the venue's withdrawal policy on behalf of myself and my co-authors.